



**Distributions of [210]Po and [210]Pb activities along the North Atlantic GEOTRACES GA01**
**(GEOVIDE) cruise: partitioning between the particulate and dissolved phase**
Yi Tang[1,2], Maxi Castrillejo[3,4], Montserrat Roca-Martí[3], Pere Masqué[3,5], Nolwenn Lemaitre[6],
Gillian Stewart[2,1]
[1] Department of Earth and Environmental Sciences, the Graduate Center, City University of New York, New York,
USA
[2] School of Earth and Environmental Sciences, Queens College, City University of New York, Flushing, USA
[3] Institut de Ciència i Tecnologia Ambientals & Departament de Física, Universitat Autònoma de Barcelona,
Bellaterra, 08193, Spain
[4] Laboratory of Ion Beam Physics, ETH-Zürich, Otto-Stern-Weg 5, Zürich, 8093, Switzerland
[5] School of Science and Centre for Marine Ecosystems Research, Edith Cowan University, Joondalup, Western
Australia, Australia
[6] Department of Earth Sciences, Institute of Geochemistry and Petrology, ETH-Zürich, Zürich, Switzerland
*Correspondence to*: Gillian Stewart (Gillian.Stewart@qc.cuny.edu)




## Abstract

Vertical distributions of total and particulate $^{210}$Po and $^{210}$Pb activities in the water column were measured at eleven stations in the North Atlantic during the GEOTRACES GA01 GEOVIDE cruise in May - June 2014. Total $^{210}$Po activity was on average 24% lower than $^{210}$Pb activity in the upper 100 m, and was closer to unity in the mesopelagic (100 – 1000 m). The partitioning coefficients ($K_d$) along the transect suggest the preferential association of $^{210}$Po relative to $^{210}$Pb onto particles. The prominent role of small particles in sorption was confirmed by the observation that over 80% of the particulate radionuclide activity was on small particles. To account for the observed surface water $^{210}$Po/$^{210}$Pb disequilibria, particulate radionuclide activities and export of both small (1-53 μm) and large (> 53 μm) particles must be considered. A comparison between the GEOVIDE total particulate $^{210}$Po/$^{210}$Pb activity ratios (AR) and the ratios in previous studies revealed a distinct geographic distribution, with lower particulate AR in the high-latitude North Atlantic (including this study) and Arctic in relation to all other samples. For the samples where apparent oxygen utilization (AOU) was calculated at the same depth and time as the $^{210}$Po/$^{210}$Pb AR (40 stations including this study), there was a two-phase correlation between the total particulate AR and AOU demonstrating the competing forces of remineralization and radionuclide decay from particles as they age.



## 1 Introduction

The major goal of the international GEOTRACES program is to characterize the distributions of trace elements and isotopes (TEIs) in the ocean on a global scale, and to identify and quantify processes that control these distributions (GEOTRACES Planning Group, 2006). The GEOVIDE section was a contribution of the French GEOTRACES program to this global survey in the North Atlantic. The GEOVIDE GA01 cruise was carried out in 2014 in the North Atlantic at latitudes greater than 40 °N and consisted of two sections: the seventh repetition of the OVIDE section from Lisbon (Portugal) to Cape Farewell (southeast tip of Greenland), and a Cape Farewell to St. John's (Canada) section across the Labrador Sea (Fig. 1). The water mass properties and main current transports have been well studied in the OVIDE section during six previous repeated hydrological surveys (2002-2012) (García-Ibáñez et al., 2015). Conditions along the Cape Farewell-St. John's section, however, were relatively unknown. The combination of the two sections constitutes a mixture of complex water masses, circulation patterns, and oceanic boundaries, presenting a special opportunity to analyze the rates of the processes that govern the distribution of TEIs.

Polonium-210 ($^{210}$Po, $T_{1/2}$ = 138.4 d) and its radioactive grandparent Lead-210 ($^{210}$Pb, $T_{1/2}$ = 22.3 y) are two non-conservative $^{238}$U decay series products. The GEOTRACES program has included both radionuclides in its TEIs list primarily due to $^{210}$Po's enhanced bioaccumulation and the use of the $^{210}$Po/$^{210}$Pb pair as a proxy for assessing particle export in the upper ocean. The distribution of $^{210}$Po and $^{210}$Pb has been widely measured over the last several decades in the Atlantic (e.g. Bacon et al., 1976), Pacific (e.g. Nozaki and Tsunogai 1976), Indian (e.g. Subha Anand et al., 2017), Arctic (e.g. Roca-Martí et al., 2016) and Southern Oceans (e.g. Friedrich and Rutgers van der Loeff 2002). However, since the data reported by Bacon et al. (1980b) at the Labrador Sea stations (47.8 – 53.7 °N), there are few studies of $^{210}$Po and $^{210}$Pb activity in the North Atlantic at latitudes greater than 40 °N. The GEOVIDE cruise, which targeted the North Atlantic from 40 °N to 60 °N, provided an opportunity to fill this data gap.

Besides ascertaining the distribution of the natural radionuclides under specific geographic conditions, this project aimed to answer questions about their biogeochemical behaviors in various marine environments. Owing to the significantly longer half-life of $^{210}$Pb relative to $^{210}$Po, the two radionuclides are expected to be in secular equilibrium (total $^{210}$Po/$^{210}$Pb activity ratio = 1) in the ocean, assuming no net removal or addition of either radionuclide. A deficit of $^{210}$Po activity relative to $^{210}$Pb activity ($^{210}$Po/$^{210}$Pb activity ratio < 1), however, is commonly found in the upper



ocean (e.g. Bacon et al., 1976; Nozaki and Tsunogai 1976; Cochran et al., 1983; Sarin et al., 1999).
This has been attributed to a higher particle reactivity of $^{210}$Po (higher partitioning coefficient, $K_d$)
than $^{210}$Pb in seawater. Particles, therefore, become enriched in $^{210}$Po ($^{210}$Po/$^{210}$Pb activity ratio >
1) and their sinking to deeper waters results in a $^{210}$Po activity deficit relative to $^{210}$Pb activity in
the upper water column where particles are formed.
In this work, we present the distributions of total and particulate $^{210}$Po and $^{210}$Pb activity at 11
stations along the GEOVIDE cruise. These data are a significant contribution to the high-latitude
North Atlantic $^{210}$Po and $^{210}$Pb activity data set. In addition, we calculate the $K_d$ of $^{210}$Po and $^{210}$Pb
during scavenging, discuss why this value has a complicated interpretation, and is mostly likely
driven by sorption to small particles. We also put our somewhat unusually low particulate
$^{210}$Po/$^{210}$Pb activity ratios (AR) into a global context and look for any possible cause of variation
along the cruise path.

**2    Methods**
**2.1 Sample collection**
The French GEOTRACES cruise to the North Atlantic (GEOVIDE, Section GA01; May 15 –
June 30, 2014) was completed on the *N/O Pourquoi Pas?*. The research vessel departed from
Lisbon, Portugal, headed northwest to the Greenland shelf, crossed the Labrador Sea, and ended
in St John's, Newfoundland, Canada (Fig. 1). A rosette equipped with conductivity-temperature-
depth sensors and 12 L Niskin bottles was used to collect 200 seawater samples (5 – 10 L each)
from 10 full water column "super" stations (16 – 22 depths/station) and 1 "Xlarge" station to 800
m (station 26, 9 depths) for the determination of total $^{210}$Po and $^{210}$Pb activity. Upon recovery,
seawater samples were transferred to 10 L acid-cleaned containers. In addition, particulate
radionuclide activities in two size classes (1-53 µm and > 53 µm) were collected at 3 – 10 depths
per station using large volume *in-situ* filtration systems (Challenger Oceanic pumps and McLane
pumps) equipped with 142 mm filter holders. Each filter head contained a stacked 53 µm PETEX
screen followed by a 1 µm pore size quartz fiber QMA filter. The volume filtered was determined
via flow meters mounted below each filter head, and the mean volume pumped through each head
was 881 L. Once recovered, clear polyethylene caps were placed on the top of the pump heads and
they were brought into a clean laboratory for sub-sampling.



**2.2 Total $^{210}$Po and $^{210}$Pb**
Total $^{210}$Po and $^{210}$Pb activities were determined from the seawater samples by the cobalt-
ammonium pyrrolidine dithiocarbamate (Co-APDC) technique (Fleer and Bacon 1984). Samples
were acidified to a pH < 2 with concentrated HCl immediately after collection and spiked with
known amounts of $^{209}$Po and stable lead as chemical yield tracers. After vigorous stirring and at
least 6 h of isotope equilibration, cobalt nitrate and APDC solutions were added to co-precipitate
Po and Pb. Samples were filtered through a 0.45 μm membrane filter and transferred into a clean
bottle, sealed with parafilm, and stored in double-bags. Further sample processing and analyses
were split between the Laboratori de Radioactivitat Ambiental (LRA) at Universitat Autònoma de
Barcelona (UAB) (samples from stations 1, 13, and 21) and the Stewart laboratory at Queens
College (QC) (stations 26, 32, 38, 44, 60, 69, and 77) to ensure higher counting statistics in the
samples. Both laboratories followed the same procedure. Briefly, the filters were digested in a
mixture of concentrated $HNO_3$ and HCl, evaporated to dryness, and eventually dissolved in 1M
and 0.5 M HCl at UAB and QC, respectively. A polished pure silver disc (Flynn 1968) with one
side covered by enamel paint was placed into the weak acid solution and heated so that the nuclides
were spontaneously plated onto only one side of the disc. The activities of both Po nuclides on the
disc were measured by alpha spectrometry. Any $^{210}$Po and $^{209}$Po remaining in the plating solution
was removed using AG 1-X8 anion exchange resin and the final solution was re-spiked with $^{209}$Po
and stored for more than 6 months to allow ingrowth of $^{210}$Po from the decay of $^{210}$Pb.
The $^{210}$Pb activity was then determined by re-plating the solutions using silver discs and
measuring the ingrown $^{210}$Po. Two aliquots of the plating solutions for each sample were taken
before the first and second platings for the measurement of total Pb concentration by inductively
coupled plasma mass spectrometry (ICP-MS) to determine sample recovery during processing.
The average recoveries produced by the LRA and Stewart groups were 83 ± 11% (n = 54) and 76
± 14% (n = 144), respectively. Finally, the initial activities of $^{210}$Po and $^{210}$Pb at the time of
collection were determined by a series of corrections, including nuclide decay, ingrowth, chemical
recoveries, detector backgrounds, and blank contamination following the methods in Rigaud et al.
(2013). The activity uncertainties from LRA were on average 8% for both $^{210}$Po and $^{210}$Pb activity,
while the activity uncertainties from the Stewart group were on average 13% for $^{210}$Po activity and
16% for $^{210}$Pb activity.



**2.3  Particulate $^{210}$Po and $^{210}$Pb**
After collection via in situ pumping, one quarter (equivalent to ~ 220 L) of the PETEX screen
containing > 53 µm or "large" particles was processed for radionuclide activity. Swimmers were
carefully removed from all samples. The QMA filters containing 1-53 µm or "small" particles
were sub-sampled (2 – 4 punches of 12 mm-diameter) achieving a mean effective volume of ~ 66
L. The screens and punches were stored in double-bags at -80 °C until the analyses onshore. The
particulate samples were split between the two laboratories in parallel to the seawater samples.
The filters were spiked with $^{209}$Po tracer solution and stable lead, digested using a mixture of
concentrated HF, $HNO_3$ and HCl at UAB, but only $HNO_3$ and HCl at QC. After multiple rounds
of digestion and evaporation to near dryness, the samples were recovered in 0.5 M HCl solution.
Any remaining pieces of filter which were not completely digested were carefully removed, rinsed
with 0.5 M HCl solution several times, and then discarded. The analyses of the particulate
radionuclide activities were identical to those for the seawater samples described in section 2.2.

**2.4 Concentration of suspended particulate matter (SPM)**
The Planquette group utilized the material on the balance of the screens and filters after
subsampling for radionuclides to determine major phase composition (particulate organic matter
(POM), lithogenic material, calcium carbonate ($CaCO_3$), opal, $Fe(OH)_3$, and $MnO_2$) (references
therein Lam et al., 2015). The complete details of sampling and analyses will be described in a
separate manuscript (Lemaitre et al., in prep.), but the mass concentration of total SPM was
calculated as the sum of the chemical dry weight of the major particulate phases.
The calculated SPM concentration was compared to the *in-situ* transmission data obtained from
the rosette CTD sensor (Fig. S1). The overall negative relationship was statistically significant ($R^2$
= 0.7, n = 53, $p$ < 0.0001), suggesting that the SPM concentrations determined were reasonable
estimates of particle concentration in the water column. We used the SPM values to determine the
partitioning coefficient, $K_d$, for $^{210}$Po and $^{210}$Pb in section 4.4.

**2.5 Primary production**
Daily primary production (PP) at each station was determined using the $^{13}$C labeling technique
by the Dehairs group. The details of sampling and analysis for PP is presented in depth elsewhere
(Fonseca-Batista et al., in review). Briefly, seawater samples (3 – 6 depths/station) were collected




from the surface to the depth of 0.2% photosynthetically active radiation (PAR). The seawater was
then incubated on deck for 24 h under conditions of photometric depths. After incubation, seawater
was filtered through GF/F filters (0.7 µm porosity), followed by $^{13}C$ determination using elemental
analysis-isotope ratio mass spectrometry. Daily PP was derived from the depth-integrated $^{13}C$
uptake rates.

**2.6 Satellite-based data**

The 8-day composites of surface chlorophyll-a concentration for each station were retrieved

from NASA's MODIS products (https://oceancolor.gsfc.nasa.gov) for the period from January to
July 2014. The time-series chlorophyll-a concentrations were used to show the development of a
phytoplankton bloom over time along the transect.

**2.7 Apparent oxygen utilization and historical values**

We compared the GEOVIDE data (particulate radionuclide activity and apparent oxygen

utilization) to historical databases and publications. The apparent oxygen utilization (AOU, µmol
kg$^{-1}$), a measurement of respiration and water mass age (Stanley et al., 2012), can be derived from
hydrological parameters (pressure, temperature, salinity, and dissolved oxygen) using the built-in
function in Ocean Data View. The location, date, database address or publication name, and type
of data (particulate $^{210}Po$ and $^{210}Pb$ activity or hydrological parameters) from all other studies is
listed in the supplemental Table S1.

**2.8  Statistical analyses**

Statistical analyses were carried out in R Studio version 3 using Fitting Linear Models, and

Welch Two Sample t-tests. Linear regression analysis was used to investigate the relationship
between total particulate $^{210}Po/^{210}Pb$ AR and AOU. The Welch Two Sample t-test was applied to
assess whether the mean of the total particulate $^{210}Po/^{210}Pb$ AR was the same as the mean of the
small particulate $^{210}Po/^{210}Pb$ AR. It was also applied to investigate the means of the total $^{210}Pb$
activity in the western and eastern sections along the transect.

**3   Results**
**3.1 Total $^{210}Po$ and $^{210}Pb$ activities**



Total $^{210}$Po activities ($^{210}$Po$_t$) in all samples ranged from 2.2 to 16.4 dpm 100 L$^{-1}$ and the mean
$^{210}$Po$_t$ for all samples was 8.8 ± 2.4 dpm 100 L$^{-1}$ (n = 198, Fig. 2). The corresponding total $^{210}$Pb
activities ($^{210}$Pb$_t$) were between 2.1 and 20.6 dpm 100L$^{-1}$ with a mean value of 10.0 ± 3.0 dpm 100
L$^{-1}$ (n = 198).
The mean $^{210}$Po$_t$/$^{210}$Pb$_t$ activity ratio (AR) of all samples was 0.92 ± 0.28 (Fig. 2, n = 198).
When considering different basins separately, there is a tendency of decreasing $^{210}$Po$_t$/$^{210}$Pb$_t$ AR
from the Western European Basin (1.10 ± 0.35) westwards to the Iceland Basin (0.90 ± 0.19) and
the Irminger Sea and the Labrador Sea (0.80 ± 0.18 and 0.83 ± 0.21, respectively). For all regions,
within the mixed layer and euphotic zone (15 – 47 m), significant deficits of $^{210}$Po$_t$ (0.80 ± 0.20, n
= 40) were observed (Fig. 3). $^{210}$Po$_t$ had enrichments below the surface at some depths at stations
1, 13, and 21 (Fig. 2) where the sub-surface $^{210}$Po$_t$ excesses were much larger than the surface
depletion. In the depth below the surface to ~ 1500 m in the Iceland Basin, the Irminger Sea, and
the Labrador Sea, the water samples still indicated a $^{210}$Po deficiency (0.84 ± 0.17, n = 27). Secular
equilibrium was generally reached near the bottom depths in all basins except at stations 13 and
60 where the water samples were enriched ($^{210}$Po$_t$/$^{210}$Pb$_t$ AR = 1.58 ± 0.16) and depleted
($^{210}$Po$_t$/$^{210}$Pb$_t$ AR = 0.50 ± 0.12) in $^{210}$Po$_t$, respectively. Secular equilibrium was also observed at
some shallow depths (i.e. 80 m at station 44) and even in surface waters (i.e. 15 m at station 38).

**3.2 Particulate $^{210}$Po and $^{210}$Pb activities**
Small particulate $^{210}$Po ($^{210}$Po$_s$) activities varied in a wide range from 0.08 to 4.82 dpm 100L$^{-1}$
(mean: 0.76 ± 0.63 dpm 100L$^{-1}$, n = 81), about 83% of the values in the small particles were lower
than 1.0 dpm 100L$^{-1}$ with higher $^{210}$Po$_s$ values generally observed in the surface samples (Table
S2). The range of small particulate $^{210}$Pb ($^{210}$Pb$_s$) activities was 0.07 to 2.89 dpm 100L$^{-1}$ (mean:
0.56 ± 0.46 dpm 100L$^{-1}$, n = 81). The vertical profiles of $^{210}$Pb$_s$ were generally similar to those of
$^{210}$Po$_s$, with relatively high activity in the surface, lower activity in the subsurface and increasing
activity with depth. This has been seen in the North Atlantic along the GEOTRACES GA03
transect (Rigaud et al., 2015). The mean $^{210}$Po$_s$/$^{210}$Pb$_s$ activity ratio (AR) was 1.43 ± 0.96 in the
surface waters (n = 14, ≤ 47 m), and 1.57 ± 0.90 with all samples included (n = 81, 8 – 3440 m).
While most surface observations had an AR of $^{210}$Po$_s$/$^{210}$Pb$_s$ higher than unity, 5 surface samples
at stations 69 and 77 showed an enrichment of $^{210}$Pb activity over $^{210}$Po ($^{210}$Po$_s$/$^{210}$Pb$_s$ AR: 0.62 ±

0.18).



Large particulate $^{210}$Po ($^{210}$Po$_l$) activities ranged from 0.01 to 0.83 dpm 100L$^{-1}$ with a mean of
0.10 ± 0.12 dpm 100L$^{-1}$ (n = 59, Table S2). The range of $^{210}$Pb activity in the large particles ($^{210}$Pb$_l$)
was from 0.02 to 0.67 dpm 100L$^{-1}$ (mean: 0.12 ± 0.14 dpm 100L$^{-1}$, n = 59). The highest $^{210}$Po$_l$ and
$^{210}$Pb$_l$ values were found at 30 m at station 26. The mean $^{210}$Po$_l$/$^{210}$Pb$_l$ activity ratio (AR) was 1.09
± 1.54 in the surface waters (n = 14, ≤ 47 m), and 1.06 ± 0.86 when all data were considered (n =
59, 8-800 m). There were 17% of the samples with a depletion of $^{210}$Po activity relative to $^{210}$Pb
activity in large particles (mean AR: 0.49 ± 0.23), particularly in surface waters from the western
section. We address this issue further in section 4.3.
The percentages of total $^{210}$Po activity in the small and large particles ranged from 0.9 to 46.7%
(mean: 8.0 ± 6.7%) and from 0.1 to 8.9% (mean: 1.2 ± 1.5%), respectively. The percentage of total
$^{210}$Pb activity ranged from 0.7 to 21.4% (mean: 4.9 ± 3.8%) and from 0.2 to 5.9% (mean: 1.1 ±
1.2%) in the small and large particulate phase, respectively. These values revealed that both
radionuclides were predominantly present in the dissolved phase along this transect, as is
commonly found in the ocean. The particulate percentages reported here are similar to the values
reported from the F.S. "Meteor" cruise 32 in the North Atlantic (Bacon et al., 1976) and along the
North Atlantic GA03 transect (Rigaud et al., 2015).
We then combined radionuclide activity on the small and large particles from the same depth
as the total particulate activity. There were 56 samples in total (surface to 800 m) and 41 of them
were from the upper 200 m. Most of the total particulate $^{210}$Po ($^{210}$Po$_p$) and $^{210}$Pb ($^{210}$Pb$_p$) activity
was on the small particles, with 86% of $^{210}$Po$_p$ and 80% of $^{210}$Pb$_p$ on the small size fraction (data
not shown). The total particulate $^{210}$Po and $^{210}$Pb AR ($^{210}$Po$_p$/$^{210}$Pb$_p$) had the same mean as that of
the small particulate $^{210}$Po and $^{210}$Pb AR ($^{210}$Po$_s$/$^{210}$Pb$_s$) (Welch Two Sample t-test, n = 56, $p$ = 0.1),
indicating that the values of the $^{210}$Po$_p$/$^{210}$Pb$_p$ activity ratios were driven by the small particles.
While the majority of particulate matter was enriched in $^{210}$Po ($^{210}$Po$_p$/$^{210}$Pb$_p$ AR > 1), there were
some surface samples that were depleted in $^{210}$Po relative to $^{210}$Pb. The $^{210}$Po$_p$/$^{210}$Pb$_p$ activity ratios
from this study are compared to the results from previous studies in various oceanic regimes in
section 4.2.

**4   Discussion**
**4.1 Total $^{210}$Po and $^{210}$Pb activities**





The overall profiles of $^{210}Po_t$ and $^{210}Pb_t$ activities were different among basins (Fig. 2). The
deficiencies of $^{210}Po_t$ activities with respect to $^{210}Pb_t$ activities in the surface samples from the
Iceland Basin, the Irminger Sea, and the Labrador Sea were generally greater than those from the
Western European Basin. Such disequilibria generally extended to the deep waters (1700 – 2950
m). In contrast, $^{210}Po_t$ activities in the Western European Basin were generally enriched relative to
$^{210}Pb_t$ activities from below the surface to the bottom of the profile. In the Western European Basin,
the sub-surface $^{210}Po_t$ activity excess was much larger than the surface depletion, suggesting that
some external source would be needed to maintain this excess $^{210}Po$ activity within the water
column. One possible source of these sub-surface $^{210}Po$ activity excesses could be the eastern
boundary upwelling along the coast of the Iberian Peninsula (García-Ibáñez et al., 2015). Even
though no strong upwelling events were revealed from temperature and density profiles during the
cruise, northerly winds favoring upwelling were recorded 2 – 3 months before the sampling
(Shelley et al., 2017). The deep water may have excess $^{210}Po$ activity due to the remineralization
of sinking particles. The upwelling of this water mass prior to the sampling date could maintain
excess $^{210}Po$ activity in the water column if the previous export of $^{210}Po$ activity was large enough.
Similar findings have been reported in the Cariaco Trench by Bacon et al. (1980a).
As atmospheric deposition is the main source of $^{210}Pb$ to the water column (e.g. Masqué et al.,
2002), we divided the GA01 transect into a western section (stn. 44 – 77) and an eastern section
(stn. 1 - 38) based on atmospheric deposition boxes described in (Shelley et al., 2017). Total
atmospheric deposition fluxes of a suite of aerosol-sourced trace metals were all reported to be
higher in the east than the west (Shelley et al., 2017). However, a two sample t-test revealed a
greater mean of $^{210}Pb_t$ activity in surface waters in the western than in the eastern section ($p < 0.02$,
mean: 12.1 vs. 10.4 dpm 100 L$^{-1}$), despite the fact that $^{210}Pb$ is usually associated with aerosols.
Even though the direct input of atmospheric $^{210}Pb$ may be larger in the east (assuming it behaves
like the other trace metals, but without aerosol $^{210}Pb$ data we cannot confirm this), alternative
inputs of $^{210}Pb$ from freshwater (e.g., sea ice processes and meteoric water) could be a greater
source of $^{210}Pb$ activity to the west. The freshwater sources over the Greenland shelf and slope
have been identified by Benetti et al. (2017), and were believed to be an important source of Fe
(Tonnard et al., in review) and Al (Menzel-Barraqueta et al., in review)  off of Greenland during
this cruise. This unexpected result highlights the need in the future to measure $^{210}Pb$ activity



simultaneously in the atmospheric and local freshwater sources in order to account for all source
terms.

**4.2 Total particulate $^{210}$Po/$^{210}$Pb AR**
A proposed explanation for the depletion of $^{210}$Po activity relative to $^{210}$Pb activity (AR <1) in
some particles is effective recycling, commonly characterized by a subsurface excess of dissolved
$^{210}$Po activity released from enriched particles leaving the surface. Bacon et al. (1976) suggested
that the efficiency of this recycling could reach up to 50%, while there is no significant concurrent
release of $^{210}$Pb activity in the water column. Laboratory studies have found the release rate of
$^{210}$Po in marine particulate matter to be significant; for example, 41% of the $^{210}$Po activity in
euphausiid fecal pellets was released over 5 days as presented in Heyraud et al. (1976). An
alternative explanation for the depletion of $^{210}$Po activity in particles is their lithogenic origin.
$^{210}$Po/$^{210}$Pb AR in lithogenic particles was reported to be similar to or less than unity (Nozaki et
al., 1998; Tateda et al., 2003). The AR < 1 observed at station 1 (120, 250, and 550 m) could be
associated with lithogenic particles from the Iberian Margin where the lithogenic contribution to
particulate and dissolved Fe and dissolved Al were reported to be significant (Gourain et al., in
review; Menzel-Barraqueta et al., in review).
The      time-series      chlorophyll-a      concentration      (8-day      composite,
https://oceancolor.gsfc.nasa.gov) from January to July 2014 at each station revealed bloom
conditions about 4 months prior to the sampling time (Fig. 4). We estimated the days since the last
bloom began prior to the sampling date for each station (Table 1) and put these data into the context
of the low $^{210}$Po$_p$/$^{210}$Pb$_p$ AR (< 1) in the total particles > 1 µm. Eight stations had total particulate
samples with $^{210}$Po$_p$/$^{210}$Pb$_p$ AR lower than unity from either shallow or deep waters. Specifically,
when the time since the last bloom began was relatively short (24 – 47 d) the samples with
$^{210}$Po$_p$/$^{210}$Pb$_p$ AR < 1 were observed in the shallow waters (10 – 60 m). In contrast, as longer time
(50 – 74 d) passed since the last bloom, the depths at which samples had $^{210}$Po$_p$/$^{210}$Pb$_p$ AR < 1 were
found to be much deeper (120 – 500 m). The results indicated that post-bloom particles could be
recycled for weeks in shallow depths and take weeks to months to sink to deeper waters.
The averages of $^{210}$Po$_p$/$^{210}$Pb$_p$ AR within the upper 200 m water column were put into a global
context with previously reported results (Fig. 5). Total particulate $^{210}$Po/$^{210}$Pb AR in the open ocean
in previous studies (e.g., Equatorial/western Pacific, Bellingshausen Sea, BATS, Labrador Sea)





were generally greater than unity. In contrast to the open ocean, the data show a distinct trend of
depletion of relative $^{210}$Po activity in marine particles from the shallow seas of the high latitude
northern hemisphere. The lowest total particulate $^{210}$Po/$^{210}$Pb AR values (Table 2, 0.4 – 0.5) were
found in the central Arctic and Chukchi shelf (Friedrich 2011; He et al., 2015). Previous studies
have observed depletion of relative $^{210}$Po activity in nearshore particles in the Yellow Sea (Hong
et al., 1999), in the turbid waters off of western Taiwan (Wei et al., 2012), on the shelf of Woods
Hole, MA (Rigaud et al., 2015), and now in the margin station off St. John's, Canada (this study).
The previous authors attributed the relative depletion of particulate $^{210}$Po activity in the nearshore
waters to the terrestrial origin/riverine input of particles with a low $^{210}$Po/$^{210}$Pb AR. This may
partially explain low activity ratios in the samples from the shelf of the Arctic Ocean as well, since
it receives ~ 10% of global river runoff and is the most riverine-influenced of all of the world's
oceans (Opsahl et al., 1999; Carmack et al., 2006). The Arctic Basin, similarly, had wide spread
deficits of particulate $^{210}$Po activity in the upper water column during the sea-ice minimum in 2007
(Roca-Marti et al., in prep.). The author suggested other particle types could also play a role in
lowering the particulate AR, including sea-ice sediments, remineralized material, fecal pellets, and
picoplankton aggregates.

**4.3 Relationship between total particulate $^{210}$Po/$^{210}$Pb AR and AOU**
Apparent oxygen utilization (AOU = $O_{2\ saturation}$ − $O_{2\ measured}$), the amount of oxygen that has
been consumed by remineralization of exported organic matter in the water column, can be used
to indicate the intensity of particle recycling (Ito et al., 2004; Duteil et al., 2013). While AOU is
generated both by water mass ageing and concomitant biological oxygen consumption (e.g. Ito et
al., 2004; Sonnerup et al., 2015), the two components of AOU would be predicted to have opposite
impacts on the $^{210}$Po$_p$/$^{210}$Pb$_p$ AR value. For example, old particles would tend to have a higher
$^{210}$Po$_p$/$^{210}$Pb$_p$ activity ratio (closer to 1) because particulate $^{210}$Po activity would increase from the
decay of $^{210}$Pb within mineral lattices and trend towards secular equilibrium ($^{210}$Po$_p$/$^{210}$Pb$_p$ AR =
1). In contrast, oxygen consumption due to bacterial remineralization would preferentially release
$^{210}$Po activity from particles into the dissolved pool (e.g. Stewart et al., 2008), leading to a lower
$^{210}$Po$_p$/$^{210}$Pb$_p$ AR in those particles.
The combination of average $^{210}$Po$_p$/$^{210}$Pb$_p$ AR and their corresponding average AOU in the
upper 200 m at 40 stations from 4 independent studies, including ARK-XXII/2 (77.38 – 87.83 ºN,





n = 15) in the Arctic, BOFS (48.89 – 49.87 ºN, n = 7), GA03 (22.38 – 39.70 ºN, n = 7), and GA01
(this study, 40.33 – 59.80 ºN, n = 11) in the North Atlantic (see map in Fig. 5) suggests two distinct
linear trends (Fig. 6). When AOU was lower than 25 µmol kg$^{-1}$, the $^{210}Po_p/^{210}Pb_p$ AR was found to
be greater than unity, together with a linear negative relationship (n = 27, $R^2$ = 0.5, $p$ < 0.001)
towards the AOU at 25 µmol kg$^{-1}$. In contrast, AOU values greater than 25 µmol kg$^{-1}$ were
coincident with a $^{210}Po_p/^{210}Pb_p$ AR < 1, and there was a linear positive relationship (n = 12, $R^2$ =
0.4, $p$ = 0.03) towards the highest AOU values measured. The two contradictory linear trends
likely reflect the opposite impacts of the two components (water mass aging and remineralization)
of AOU on $^{210}Po_p/^{210}Pb_p$ AR. This suggests that the variation in the $^{210}Po_p/^{210}Pb_p$ AR was mainly
driven by remineralization processes under the condition of AOU < 25 µmol kg$^{-1}$, lowering the
total particulate activity ratio; whereas the decay of $^{210}Pb$ into $^{210}Po$ towards secular equilibrium
may dominate when AOU was > 25 µmol kg$^{-1}$, leading to an increase in $^{210}Po_p/^{210}Pb_p$ AR. This
explanation, however, appears to only hold for the high latitude Northern Hemisphere where
$^{210}Po_p/^{210}Pb_p$ activity ratios were generally lower than those in the other oceanic settings (Fig. 5).
In the high latitude Southern Hemisphere near Antarctic (e.g., ANT-X/6), for example, there is no
apparent relationship between $^{210}Po_p/^{210}Pb_p$ activity ratios and AOU. This relationship (or lack
thereof) deserves more study in the future.

**4.4 Small particles, sorption, and calculating POC export**

The assumption that the largest particles dominate export in the ocean (e.g. Bishop et al., 1977;

Fowler and Knauer 1986; Michaels and Silver 1988; Honjo et al., 1992; Walsh and Gardner 1992)
has been challenged by increasing studies which argue that small particles can form aggregates
that sink, and their contribution to carbon export could be larger than previously thought (e.g.
Richardson and Jackson 2007; Lomas and Moran 2011; Amacher et al., 2013; Puigcorbé et al.,

2015).

We investigated the role of small phytoplankton to carbon export along the GA01 transect via

investigation of pigments and *in-situ* primary production. The fraction of pigment-based size
classes suggested a significant contribution of small particles (nano-phytoplankton: 2 – 20 µm
60%, pico-phytoplankton: < 2 µm, 13%) to primary production in the eastern section while larger
particles (micro-phytoplankton: > 20 µm, 60%) may have dominated production in the western
section of the GA01 transect (Tonnard et al., in prep.). The rate of primary production in the eastern



section (mean: 99 ± 50 mmol C m$^{-2}$ d$^{-1}$), however, was similar to that in the west (mean: 93 ± 58
mmol C m$^{-2}$ d$^{-1}$) (data not shown). While we do not have direct evidence of small particles sinking,
we are making an assumption that our study sites behave as the above cited papers have seen
elsewhere. Therefore, a possible link between small particles and production, and possibly export
(proportional to their role in production according to Richardson and Jackson, 2007), may exist
along the transect.
The partitioning coefficient, $K_d$ (L kg$^{-1}$), has been used to describe the particle adsorption
behavior of radionuclides. It is defined as the ratio of the adsorbed radionuclide activity ($A_p$, dpm
100L$^{-1}$) to the dissolved radionuclide activity ($A_d$, dpm 100L$^{-1}$), normalized by the suspended
particulate matter concentration ($SPM$, µg L$^{-1}$):
$$K_d = \frac{A_p}{A_d} \times \frac{1}{SPM} 10^9 \qquad (1)$$
Owing to the different biological and chemical behaviors of $^{210}$Po and $^{210}$Pb, the interpretation
of measured $K_d$ for $^{210}$Po ($K_d$(Po)) may not be as clear as that for $^{210}$Pb ($K_d$(Pb)) (i.e. $K_d$(Po) also
takes the fraction of absorbed $^{210}$Po into account, Tang et al., 2017). As such, it would be more
appropriate to think of both $K_d$(Po) and $K_d$(Pb) as the intensity parameter for the radionuclide
association with particles.
In this study, the size-fractionated data of radionuclide activity and SPM allowed us to
calculate the partitioning coefficients for both radionuclides on small and large particles. We
present only the coefficients for the small particulate phases ($K_d$(Po)$_s$, $K_d$(Pb)$_s$) and the total
particulate phases ($K_d$(Po)$_p$, $K_d$(Pb)$_p$) because most of the particulate activity (> 80%) was
associated with the small particles along the GEOVIDE transect, and most conceptualized
scavenging models consider either the two-box model (dissolved – total particulate phases, i.e.
$K_d$(Po)$_p$) or the three-box model (dissolved – small – large, i.e. $K_d$(Po)$_s$) (Clegg and Whitfield 1990;
1991; Rigaud et al., 2015) and thus activity is concentrated from the dissolved phase to the total
or small particles. The $K_d$ values for the small particulate phase were slightly higher than those for
the total particulate phase but overall these values were very similar for both radionuclides (Fig.
7). Combining the fact that adsorption/scavenging was in fact driven by small particles with the
contribution of small phytoplankton to production, the importance of small particles to
radionuclide export is suggested. We recommend combining the activities of both small and large
particles into a total particulate fraction in order to explain total $^{210}$Po/$^{210}$Pb disequilibria in the




surface waters, and utilizing the characteristics of the total particles (instead of just the large
particles) in the estimation of the POC export fluxes (Tang et al., in prep.).

**5    Conclusions**

In this study, we reported the vertical distribution of total and size-fractionated particulate $^{210}$Po

and $^{210}$Pb activities in the North Atlantic during the GEOVIDE GA01 cruise. More than 90% of
the radionuclide activity was found in the dissolved phase, while a small proportion was associated
with particles in this transect. Total $^{210}$Po activity was generally depleted relative to total $^{210}$Pb
activity in the upper 100 m due to the assumed preferential adsorption and uptake of $^{210}$Po activity
by particles.

Over 80% of the particulate radionuclide activity was on small particles, and it appeared that

the adsorption/scavenging of both radionuclides was driven by small particles. Considering this
and the contributions of small phytoplankton to primary production (and possibly export), we
suggest combining the activities of both $^{210}$Po and $^{210}$Pb from both small and large particles into a
total particulate fraction (> 1 μm) in order to explain the water column $^{210}$Po/$^{210}$Pb disequilibria
and calculate POC export.

There appear to be geographic differences in particulate $^{210}$Po/$^{210}$Pb activity ratios measured

during GEOVIDE and previous studies, with particularly low values in the high-latitude North
Atlantic and Arctic. While this observation deserves more attention, we support previous
suggestions that this is due to the terrestrial origin/riverine input of particles with a low $^{210}$Po/$^{210}$Pb
AR into the river-dominated shallow basins of the Arctic. Considering the age of the particles and
water masses as well as the importance of remineralization may also explain some of these
observations, as there was a significant relationship between the total particulate activity ratio and
AOU when both were measured in the high latitude North Atlantic and Arctic Oceans.


**Acknowledgements**

Thank you to the chief scientists (G. Sarthou and P. Lherminier) of the GEOVIDE cruise, and the
captain (G. Ferrand), and crew of the *N/O Pourquoi Pas?* for their support of this work. Many
thanks to P. Branellec, F. Desprez de Gésincourt, M. Hamon, C. Kermabon, P. Le Bot, S. Leizour,




O. Ménage, F. Pérault, and E. de Saint-Léger for their technical support during the GEOVIDE
expedition, and to C. Schmechtig for the GEOVIDE database management. P. Lam is also
acknowledged for providing two modified McLane ISP. Special thanks go to the member of the
pump group including F. Planchon, V. Sanial, and C. Jeandel. The author would like to thank C.
Mariez, S. Roig, F. Planchon, and H. Planquette who helped in providing particle composition data
and A. Roukaerts, D. Fonseca-Batista, F. Deman, and F. Dehairs for primary production data. We
also would like to acknowledge the funding agencies: the French National Research Agency
(ANR-13-BS06-0014, ANR-12-PDOC-0025-01), the French National Center for Scientific
Research (CNRS-LEFE-CYBER), the LabexMER (anr-10-LABX-19), and Ifremer. G. Stewart
and Y. Tang were supported by NSF award #OCE 1237108. M. Castrillejo and M. Roca-Marti
were funded by an FPU PhD studentship (AP-2012-2901 and AP2010-2510, respectively) from
the Ministerio de Educación, Cultura y Deporte of Spain. M. Castrillejo was also supported by the
ETH Zurich Postdoctoral Fellowship Program (17-2 FEL-30), co-funded by the Marie Curie
Actions for People COFUND Program. Additional thanks go to G. Hemming and T. Rasbury for
laboratory assistance with the ICP-MS analyses.




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




Table 1. Biological characteristics of the water column determined by chlorophyll-a
concentration (8-day composite) from Fig. 4, including the date when the last bloom began,
the difference in chlorophyll-a concentration between the sampling time and last bloom peak,
and the days since the last bloom. Activity ratios of $^{210}Po_p/^{210}Pb_p < 1$ and their corresponding
depths are also shown. *NA* indicates that all samples from the corresponding depth range had
$^{210}Po_p/^{210}Pb_p$ equal to or greater than 1 (no sample with $^{210}Po_p/^{210}Pb_p < 1$).

| Station | Sampling date | The date last bloom began | Last bloom peak-current state | Days since last bloom | $^{210}Po_p/^{210}Pb_p < 1$ | |
|---|---|---|---|---|---|---|
| | | | | | 0-100 m | > 100 m |
| 1 | 5/19/14 | 3/6/14 | Large | 74 | *NA* | Yes (120, 250, 500 m) |
| 13 | 5/24/14 | 4/7/14 | Small | 47 | Yes (60 m) | *NA* |
| 21 | 5/31/14 | 4/7/14 | Large | 54 | *NA* | Yes (120 m) |
| 26 | 6/4/14 | 4/15/14 | Large | 50 | *NA* | Yes (400 m) |
| 32 | 6/7/14 | 5/9/14 | Small | 29 | *NA* | *NA* |
| 38 | 6/10/14 | 5/17/14 | Small | 24 | Yes (60 m) | *NA* |
| 44 | 6/13/14 | 5/9/14 | Small | 35 | *NA* | *NA* |
| 60 | 6/18/14 | 5/17/14 | Large | 32 | *NA* | *NA* |
| 64 | 6/19/14 | 5/17/14 | Small | 33 | Yes (30 m) | *NA* |
| 69 | 6/22/14 | 5/25/14 | Small | 28 | Yes (20, 30 m) | *NA* |
| 77 | 6/26/14 | 5/25/14 | Small | 32 | Yes (10, 20, 50 m) | *NA* |







Table 2. Global compilation of total particulate $^{210}Po/^{210}Pb$ activity ratios ($^{210}Po_p/^{210}Pb_p$) in the upper 200 m including this study.

| Region | | Sampling Method | Date | Size (μm) | Depth (m) | $^{210}Po_p/^{210}Pb_p$ | Reference |
|---|---|---|---|---|---|---|---|
| Arctic | CESAR | In-situ pump | Apr – May 83 | > 0.45 | 2-200 | 1.2 | (Moore and Smith 1986) |
| | Arctic (ARK-XXII/2) | Niskin bottle | Jul-Sep 07 | > 0.45 | 10-200 | 0.5 | (Friedrich 2011) |
| | Chukchi Shelf | Niskin bottle | Jul-Sep 10 | > 0.45 | 0-90 | 0.4 | (He et al., 2015) |
| Atlantic | F.S. Meteor | Niskin bottle | Nov-Dec 73 | > 0.4 | 0-200 | 3.1 | (Bacon 1977) |
| | Cariaco Trench | Niskin bottle | Dec 73 | > 0.4 | 0-200 | 1.8 | (Bacon et al., 1980a) |
| | Labrador (R/V Knorr) | Niskin bottle | Jun 75 | > 0.4 | 0-100 | 3.9 | (Bacon et al., 1980b) |
| | South of New England | Niskin bottle | Jul 80 | > 0.45 | 4-200 | 1.8 | (Bacon et al., 1988) |
| | N. Atlantic (BOFS) | Niskin bottle | May-Jun 89, 90 | > 0.45 | 0-150 | 5.1 | (BODC et al., 2016) |
| | South-equa. Atlantic | Niskin bottle | May-Jun 96 | > 0.7 | 10-200 | 1.7 | (Sarin et al., 1999) |
| | BATS | Go-Flo bottle | Oct 96 | > 0.45 | 0-200 | 3.7 | (Kim and Church 2001) |
| | N. Atlantic (GA03) | In-situ pump | Oct-Nov 10, Nov-Dec 11 | > 0.8 | 30-200 | 1.5 | (Rigaud et al., 2015) |
| | N. Atlantic (GA01) | In-situ pump | May-Jun 14 | > 1 | 8-200 | 1.4 | This study |
| Pacific | North Pacific | Niskin bottle | Nov 73 | > 0.4 | 10-150 | 8.5 | (Bacon et al., 1976) |
| | W. Pacific (FR05/92) | Niskin bottle | Jul 92 | > 0.45 | 0-200 | 1.3 | (Towler 2003) |
| | Equa. Pacific | Go-Flo bottle | Aug-Sept 92 | > 0.45 or 0.5 | 0-200 | 5.1 | (Murray et al., 2005) |
| | W. Pacific (FR08/93) | Niskin bottle | Nov 93 | > 0.45 | 0-200 | 15.7 | (Towler 2013) |
| | W. Pacific (FR07/97) | Niskin bottle | Aug 97 | > 0.45 | 0-200 | 7.2 | (Peck and Smith 2002) |
| | Aleutian Basin | Niskin bottle | Jul-Aug 08 | > 0.2 | 0-200 | 1.9 | (Hu et al., 2014) |
| | E. Pacific (GP16) | In-situ pump | Oct-Dec 13 | > 1 | 15-200 | 2.4 | unpublished |
| Antarctic | S. Ocean (ANT-X/6) | Niskin bottle | Oct-Nov 92 | > 0.45 | 20-200 | 3 | (Smetacek et al., 1997) |
| | Bellingshausen Sea | Go-Flo bottle | Nov-Dec 92 | > 0.45 | 0-100 | 13.9 | (Shimmield et al., 1995) |
| | S. Ocean (ANT-XXIV/3) | Niskin bottle | Feb – Apr 08 | > 0.45 | 25-200 | 1.3 | (Friedrich et al., 2011) |
| | S. China Sea | Go-Flo bottle | Jan-Oct 07, May 08 | > 0.45 | 0-200 | 1.7 | (Wei et al., 2014) |





| | | | | | | |
|---|---|---|---|---|---|---|
| Margin Sea | W. Taiwan | Go-Flo bottle | Apr 07 | > 0.45 | 8-25 | 0.8 | (Wei et al., 2012) |
| | Yellow Sea | Niskin bottle | Feb 93 | > 0.7 | 0-100 | 0.9 | (Hong et al., 1999) |
| | Mediterranean Sea | Sediment trap | Mar-Jun 03 | | 200 | 4.5 | (Stewart et al., 2007) |



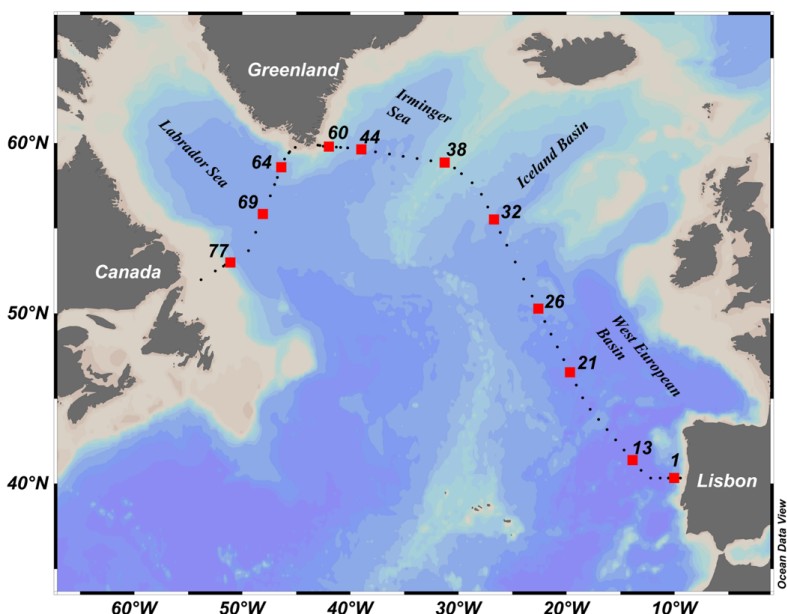


Fig. 1. Map of the GEOVIDE cruise track (black dots) and the 11 stations sampled for ${}^{210}$Po and
${}^{210}$Pb activity (red squares). Each sampling location is labeled with a station number. The
sampling stations are divided into 4 regions (from east to west): West European Basin (stations
1, 13, 21, 26), Iceland Basin (stations 32, 38), Irminger Sea (stations 44, 60), and Labrador Sea
(stations 64, 69, 77).



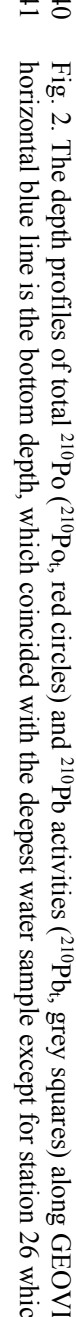

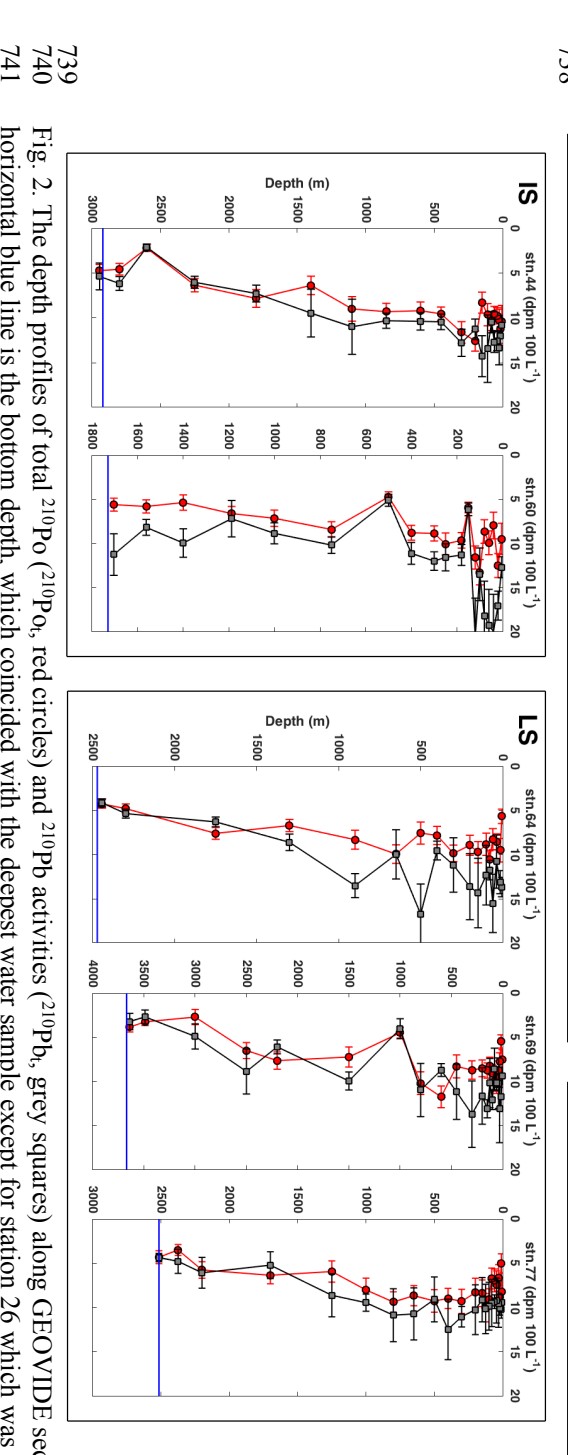

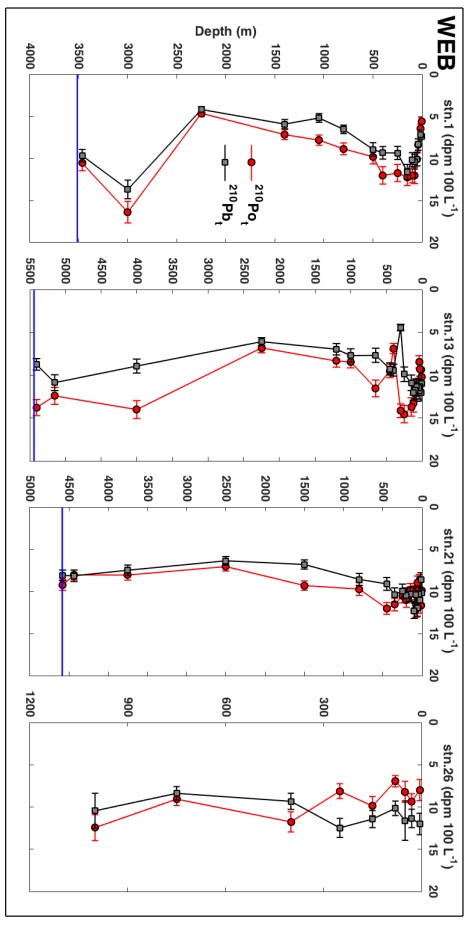

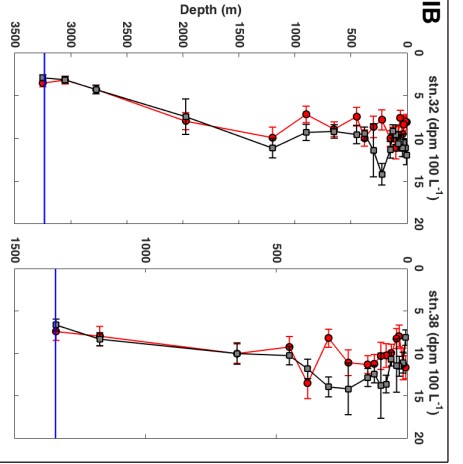

Fig. 2. The depth profiles of total $^{210}$Po ($^{210}$Po$_t$, red circles) and $^{210}$Pb activities ($^{210}$Pb$_t$, grey squares) along GEOVIDE section. The horizontal blue line is the bottom depth, which coincided with the deepest water sample except for station 26 which was sampled only





down to 1000 m. Note that the depth scale for each plot may be different. The profiles are shown in the order of sampling date with
the region indicated on the top left of each box: Western European Basin (WEB), Iceland Basin (IB), Irminger Sea (IS), Labrador Sea
(LS).








Fig. 3. A closer look at only the zoom for the upper 250 m of the depth profiles of total $^{210}$Po ($^{210}$Po$_t$, red circles) and $^{210}$Pb activities ($^{210}$Pb$_t$, grey squares) along the GEOVIDE section. The horizontal orange and magenta lines denote the mixed layer depth (MLD) and





the base of the euphotic zone ($Z_{1\%}$), respectively. The depth profiles are shown in the order of sampling and grouped by region (refer
to Fig. 2 for the text abbreviations).



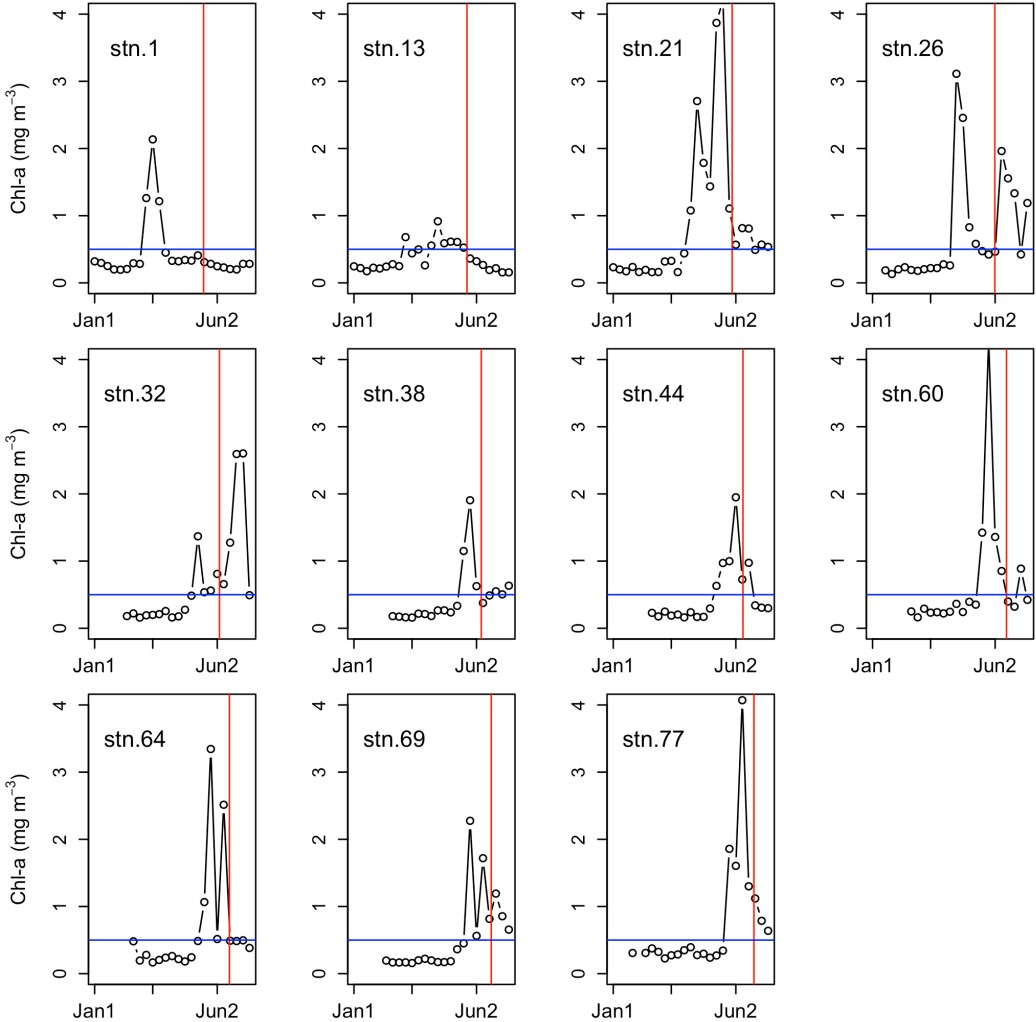


Fig. 4. Time-series (January 1 – July 12, 2014) chlorophyll-a concentrations (8-day
averages) from Aqua MODIS (https://oceancolor.gsfc.nasa.gov) at each station along
the GA01 transect. The vertical red line denotes the sampling date at each station. The
horizontal blue line denotes chlorophyll-a concentration of 0.5 mg m$^{-3}$. The time
when chlorophyll-a concentration first exceeded 0.5 mg m$^{-3}$ after the end of the last
bloom defines the date the next bloom began.



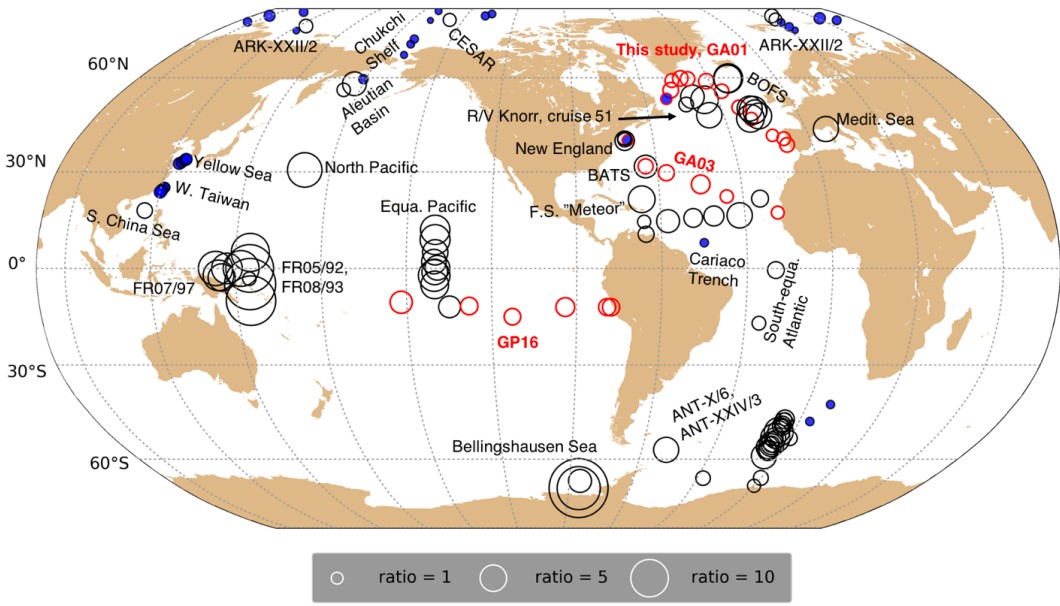


Fig. 5. Comparison of particulate $^{210}$Po/$^{210}$Pb activity ratios in the upper 200 m from this
study and 20 previous studies (references in Table 2). Information about the study site,
sampling date, method, and particle size of each study are shown in Table 2. The black
circles represent data from previous studies while the red circles are the results from samples
analyzed in the Stewart lab from three recent GEOTRACES transects (GA03, GP16, and this
study, GA01 GEOVIDE). The filled blue and open circles indicate activity ratios lower and
higher than 1, respectively.



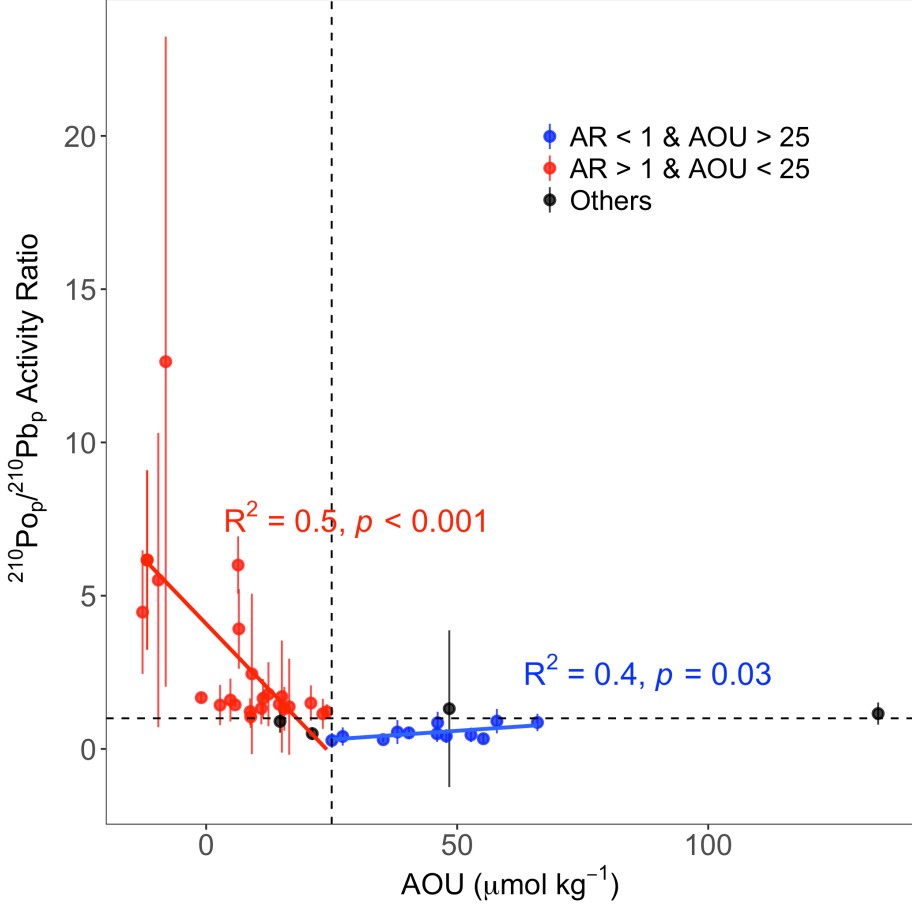

767

Fig. 6. The relationship between AOU ($\mu$mol kg$^{-1}$) and total particulate $^{210}$Po/$^{210}$Pb

activity ratio ($^{210}$Po$_p$/$^{210}$Pb$_p$) from the upper 200 m in the northern hemisphere (> 22 ºN)

investigated by a linear regression model (red and blue lines). The 40 stations include

data from previous studies, ARK-XXII/2 (77.38-87.83 ºN, n = 15) in the Arctic, BOFS

(48.89-49.87 ºN, n = 7), GA03 (22.38-39.70 ºN, n = 7), and this study, GA01 (40.33-

59.80 ºN, n = 11) in the North Atlantic. The horizontal dashed line represents

$^{210}$Po$_p$/$^{210}$Pb$_p$ AR = 1 and the vertical dashed line represents AOU = 25 $\mu$mol kg$^{-1}$. Red

circles denote the average $^{210}$Po$_p$/$^{210}$Pb$_p$ >1 and AOU < 25 $\mu$mol kg$^{-1}$, while blue circles

denote the average $^{210}$Po$_p$/$^{210}$Pb$_p$ < 1 and AOU > 25 $\mu$mol kg$^{-1}$. Data that are in neither

category are denoted by the black circles.





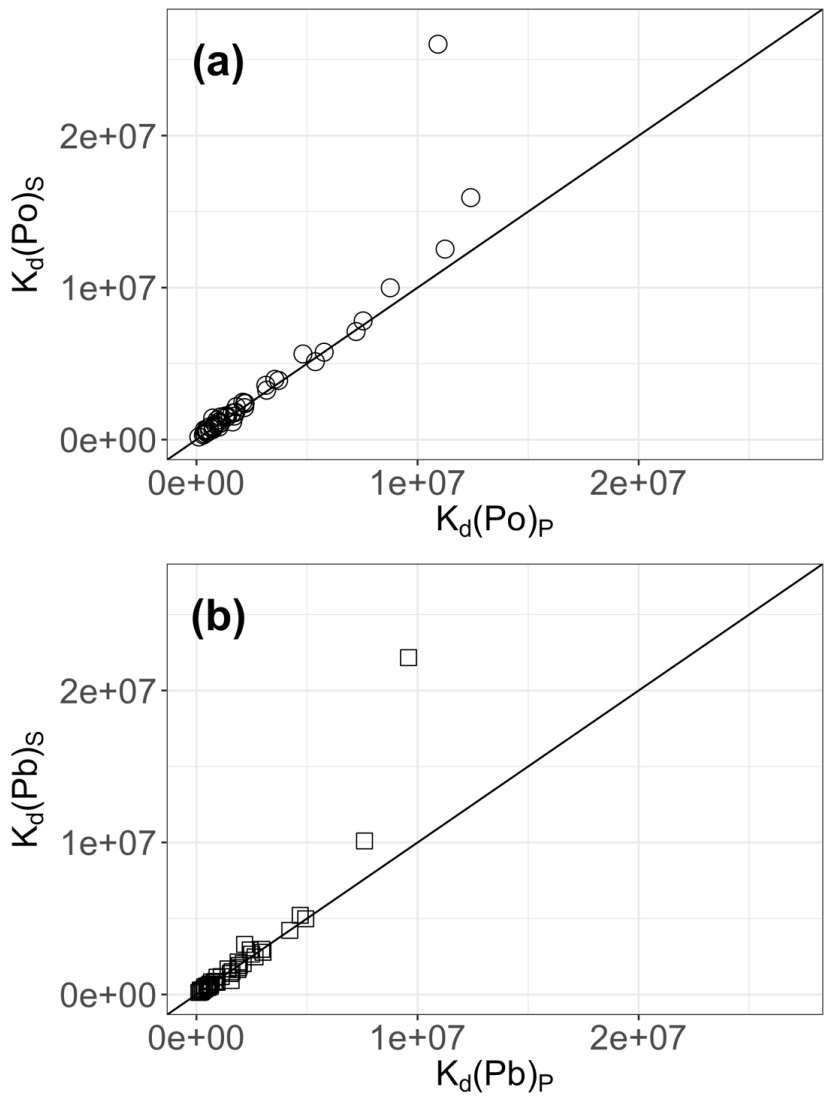


Fig. 7. Comparison of the partitioning coefficient ($K_d$) between the dissolved and small
particulate phases ($K_{ds}$) vs. between the dissolved and total particulate phases ($K_{dp}$) for (a)
$^{210}$Po and (b) $^{210}$Pb. The 1:1 line is indicated as the solid line in each plot.