# Peer review of "Distributions of total and size-fractionated particulate 210Po and 210Pb activities along the 1 North Atlantic GEOTRACES GA01 (GEOVIDE) transect 2 3 Yi Tang1,2, Maxi Castrillejo3,4, Montserrat Roca-Martí3, Pere Masqué3,5, Nolwenn"

_Biogeosciences, 2018_

## Referee Comment (RC1) · Anonymous Referee #1 · 18 Jun 2018

Review for Biogeosciences (bg-2018-210) Authors: Tang, Castrillejoi, Roca-Marti, Masque, Lemaitre and Stewart Title: "Distribution of 210-Po and 210-activities along the North Atlantic GEOTRACES GA01 cruise"

General Comments: This paper presents valuable data of high quality in an important region of the far North Atlantic. Previous data over the past decades since GEOSECS and other expeditions are presented in comparison. However much of this older data lacked the high resolution in space and time, including both dissolved and particulate samples in two size fractions. Thus provided is how the nuclide distribution is im-

pacted over the course of the cruise track period. The data is treated primarily in a statistical oceanographic manner that lends insight into how this temporally sensitive daughter/grandparent nuclide pair compare to biogenic parameters such as satellite chlorophyll and apparent oxygen utilization. As such it provides a historical context on the conditions that proceeded the cruise while the couple relaxed over the previous months.

Specific Comments: One unfortunate aspect of the paper is that it fails to model the data in the context of biogenic carbon flux, the primary strength of the nuclide pair. Perhaps the organic carbon data are missing, or awaiting a more complete synthesis with other nuclides such as 234-Th, as done admirably before by the UAB lab group.

Technical Issues:

1 Introduction It is noted that there is significant benthic disequilibrium (210-Po deficiency) well below the euphotic zone, indeed significantly below the main thermocline at times (e.g. 4000 meters at Station 13; 1400 meters at station 60). This dilemma and benthic consequences has been discussed in the recent literature (Rigaud, et al., 2014). Page 3: As such, maybe the literature citations in the introduction that need to be updated for the current millennium!

2 Methods Page 4: What is meant by "Xlarge" station (26), as the number of depths are less than others? Page 5: Is six hours sufficient for equilibration, or were there previous tests performed to verify this?' What was the time lag between sample processing on board, and nuclide separation in the lab on shore, unless both were done on board? This can be important as reviewed in Rigaud, et al. 2013. Evidently this is reflected in the data reported in supplemental tables, although there are not errors assigned to the nuclide ratios in Table 2. Page 6: Who are the "Planquette group"?

3 Results Page 7: As noted above, stations 13 and 30 appear to have total 210-Po deficiency at depth (Fig. 2), not excess. Page 8: Increase in activity with depth for both nuclides is not evident in Figs. 2 and 3, rather decrease.

4 Discussion Page 10: Usually in the far North Atlantic, 210-Pb association with aerosol dust is not as evident in the east, rather alternative fresh water sources (e.g. precipitation) as noted in the west. Page 11. The lithogenic source of a depleted 210-Po/210-Pb ratio should only be evident if the atmospheric scavenging was in the form of precipitation. Alternatively or as with lithogenic particles from the continental margin, the 210-Po has been preferentially extracted lately in fecal pellets by organisms. Page 12: The alternative scenario is noted here at the end of section 4.2. As such, might there be a corresponding dissolved ration greater than one? Page 13: The negative relationship between AOU and 210-Po/210-Pb is not very strong. Page 14: Line 387 appears not to be clearly expressed indeed!

5 Conclusion Page 15: The impact of a terrestrial origin on the 210-Po/210Pb ratio less that unity might indeed be born out in the Arctic basin during summer seasons of strong biogenic processing. Maybe there is evidence in the recent GEOTRACES cruises on time scales of several months conclusive with that of the grand-daughter/parent nuclide pair?

Figure Captions 4) . . .bloom defines the date when the next bloom began. 5) The black and blue colored circles are not well distinguished.

---

## Referee Comment (RC2) · Anonymous Referee #2 · 9 Jul 2018

**Reviewer Recommendation and Comments for Manuscript BG-2018-210**

**General comments**

The manuscript reports on total, small particles and large particles activity of 210Po and 210Pb nuclides along the North Atlantic GEOTRACES GA01 (GEOVIDE) cruise. The paper is well written, well structured and I believe that such measurements in this areas are essential for the scientific understanding of TEI's and biogenic elements in the global ocean. The approach is very good and the compilation of many other joined data (AOU, PP, SPM, chlorophyll, ...) is essential to reach this goal. In addition there is a huge effort to include this new dataset with previous ones in order to get a better view of the processes controlling the behaviors of 210Po and 210Pb at a larger scale. Finally I found very interesting news findings that emerge from a new way to confront this 210Po and 210Pb dataset to other variables (comparison with chlorophyll-a from satellite-based data, AOU, ...) that merit to be published.

However I found some questioning points that need to be addressed:

- the splitting of the samples between two different labs with two methods that differ in some points is very surprising. Some practical reasons can certainly explain this procedure but they are not mentioned. The reader need to be sure that the results can be compared. Especially since there are distinct features that can be seen between the samples from the two labs and that a part of the discussion relies on such differences.

- the last section of the discussion about the sorption, distribution coefficient and implication for particles and POC export very speculative. This is embarrassing as this appears in the abstract and the conclusion as the most important finding of the study while there are other findings much more robust that are not presented in that way.

- the presentation of the context in the introduction and the state of the art about 210Po and 210Pb isotopes in the ocean is a little bit weak and I think the importance of such measurement in this area should be specifically strengthened.

Consequently, I believe this paper must be published when these points will be addressed.

**Specific comments:**

Title:

- The part of the title "partitioning between the dissolved and particles phase" is maybe not really appropriate as the most important discussions in the paper is about the processes explaining the variations in the 210Po/210Pb activity ratio within each phase (i.e., total and small/large particles).

Abstract

P2, L22-23: this was not shown in the manuscript

Introduction

P3, L42: "seventh repetition of the OVIDE section": please precise what is the OVIDE section/program?

P3, L44-46: please give a short summary on the hydrological properties on the area.

P3, L47-49: you should illustrate what is this expected "mixture of complex ..." and why this section may present a special opportunity.

P4, L72-78: I found this objectives section disappointing and clearly not ambitious enough with respect to the dataset compiled and presented in this paper. I suggest the authors to strengthen this part.

Methods

P5, L104: Please correct the sentence to avoid confusion: what was transferred into a clean bottle? The filters? The filtrate?

P5, L107: is the "Stewart laboratory" the official name of the laboratory?

P5, L107-108: why this splitting procedure of the sample? The reader need to know why this

splitting procedure allow to "ensure higher counting statistic in the samples". Did the laboratory performed intercalibration experiments?

P5, L110: Please correct the sentence to avoid confusion: the filter was not evaporated to dryness.

P5, L110: Remove "eventually"

P5, L112: what weak acid solution?

P5, L120: write "to determine Pb recovery" instead of "to determine sample recovery".

P5, L125-127: why this difference between the two labs?

P6, L137: These two different digestion procedures may give different results? Please explain if tests were carried out. Are the data from the two groups comparables?

P6, L144: what is the Planquette group?

P6, L144: is this sentence correct: "the material on the balance of the screens and filters"?

P6, L148-149: if the method is the same as described by Lam et al. 2015, I suggest to remove the Lemaitre et al. in prep. reference if it is not published at the time of the publication of this paper. Same comment for other reference in prep. in the manuscript.

P6, L158: what is the Dehairs group?

P6, L157-164: a little bit more details is needed here: how the photometric conditions was applied on deck? I guess that 13C was spike before the incubation? ...

P7, L173-174: Before to compare the AOU data from the GEOVIDE program, you should explain how you get it. In facts, the section 2.7 is disturbing. There are two things here: the AOU and the comparison with historical data but there is no link between them. I suggest to split this section in two (even short) sections.

P2, L182-187: SPM, PP, chlorophyll were not considered to try to explain the 201Po-210Pb activities and activity ratios distribution?

Results

- p7, L195-202: there are a clear difference between station 1, 13, 21 and the other ones. These differences also correspond to the two samples groups that were processed by two labs. This is embarrassing if there is nothing that certify that labs results can be compared.

- p7, L200-202: please rewrite this sentence which is very confusing.

- p7, L195-207: this paragraph is confusing. Please describe firstly the surface water then the depth (or in the other way) but not a mixing description.

- p8, L214-216: why the figure is not shown? The particulate profiles should be plotted (at least in the appendix material).

- p9, L242-244: yes, this is not surprising as the small particle are the main particulate reservoir.

- p9, L245-246: which particulate samples are depleted? Where they are located? In surface? Subsurface? Variable depths?

Discussion

- p10, L264-265: large excess is not seen at depth.

- p10, L260-267: I don't understand how an upwelling along the Iberian coast can bring excess 210Po all over the water column in the 3 station from the WEB.

- p11, L295-298: what do you mean by significant? Are they significantly different than this other station? Statistically tested? Is this confirmed from the data on the geochemical composition of SPM?

- p11, L304-308: this is an interesting point. Is there a figure (or a way) to illustrate this? For example a plot showing the AR in surface or subsurface as a function of the time since the last bloom?

- p12, L321: Is this particulate 210Po depletion in the coastal sea related to the 210Po/210Pb AR in these the terrestrial/riverine particles or is this due to the nature of those particles that present a lower scavenging efficiency of dissolved 210Po with respect to 201Pb?

- p12, L331: AOU must be defined in the method section. What a negative AOU value means?

- p12, L332: remineralization + respiration + oxidation reactions.

- p12, L333-334: I do not see why water mass aging may change the OAU if there is no mineralization. To my opinion, only biogeochemical processes may change OAU values while the

time can only change the intensity of O2 consumption by those biogeochemical processes. I think this should be better specified in this part to avoid confusion.
- p12, L336: what is an old particle? Weeks? Months? Years?
- p12, L336-338: time will induce an AR approaching 1: decreasing AR if the initial AR is >1 and increasing if the initial AR is < 1. Here you hypothesis that the initial AR in particle is <1 but both cases are possible. Please correct.
- p12, L343-357: very interesting results and interpretation! However, I have two mains questions:
    - Why the increase of AR from negative value to value close to 1 for OAU > 25 µmol/kg? Higher the OAU, higher the mineralization. So intuitively, the AR should be maintained more and more negative with increasing OAU?
    - I do not understand why it is said that this observation stands only for high latitude in the northern hemisphere. Other campaigns from high latitude in the Northern hemisphere are also reported on figure 5 but are not considered. In addition, GA-03 campaign are not from high latitude. What gives this relationship for other campaigns? Why this 4 campaigns was selected?
- p13, L370: What do you mean by investigation of pigments? There is nothing about it in the material and methods section.
- p14, L377-378: what do you mean by "as the above cited papers have seen elsewhere"? Please precise
- p14, L378-380: this is expected for the eastern part of the transect only?
- p14, L391-392: how did you calculate the dissolved activity? This is not indicated. When you consider the Kd for the small particles you normalize with the SPM for the small particles also? Same question for the total particulate. Please precise.
- p14, L399-401: How this is possible as the small particulate activity is necessary lower than the total particulate activity? Is it associated to the SPM normalization?
- p14, L401-403: here you affirm that the scavenging and export is mostly driven by small particles. But there is nothing to confirm this. Although this can be plausible, this is just an hypothesis.
- p12-14, L362-404: this section is very surprising. From the title of the section I excepted to find POC export calculation. In facts, there is no data really discussed or even showed (pigment, primary production, ...) and most of the discussion is based on hypothesis without real solid basis to support them. I suggest to rewrite this section around concrete data only and to change the title of this section.

Conclusion:
- p15, L415-420: again this was not clearly demonstrated. This conclusion should be very robust because it can have large implications in the future sampling strategy. Differently: does the sampling and analysis of two particulate size fractions is necessary in the future? So this has to be very robustly demonstrated. I agree with the fact that the high proportion of particulate nuclides is found in the small particle indicates that small particles are important in the sorption process. But I'm clearly not convinced from the data showed in the manuscript there is evidence to say that the small particles play an important role in the export of particles. If so, this should be strengthened.
- I may suggest to synthesis the most important findings based on the data only. There is nothing on the time elapsed since the last bloom for example.

Fig 3:
- I doubt the sentence "A closer look at only the zoom" is correct in the caption
- Stn 60: 2 dot are missing for 210Pb at approximatively 50 m and 120 m depth.

Figure 6:
- negative AOU value need to be explained?
- with the uncertainty on Po/Pb AR there is (most of the time) not significant deviation from the 1 AR for the "other points". I suggest to integrate the "other points" within the regression keeping the only separation lower or above 25 µmol/kg for OAU.

Figure 7:
- the axis labels on the figure and in the caption are not the same. Please homegeneize.

---

## Author Comment (AC1) · 30 Jul 2018

Response to Referee 1

We thank Referee 1 for the helpful comments. We will address all changes in the revised manuscript as detailed in our responses below. The referee comments are in black and their line numbers refer to the original submitted manuscript. Our responses are in blue text.

We want to note that some of the reviewer's comments may pertain to an original draft of the manuscript which has already been revised. We have tried to address all comments, but in some cases, we do not see what the reviewer is talking about. In our response, we will only be referring to the version that is currently available on the BG website.

We also want to note that we now have submitted a companion paper to this special issue that is specifically about the particulate organic carbon (POC) export using the $^{210}Po/^{210}Pb$ technique.

- Specific Comments: One unfortunate aspect of the paper is that it fails to model the data in the context of biogenic carbon flux, the primary strength of the nuclide pair. Perhaps the organic carbon data are missing, or awaiting a more complete synthesis with other nuclides such as 234-Th, as done admirably before by the UAB lab group.

We understand the reviewer's concern. In fact, we have submitted two manuscripts to this special issue. In the manuscript reviewed here we discuss the general distribution of $^{210}Po$ and $^{210}Pb$ activity along the GEOVIDE transect. The second manuscript entitled "The export flux of particulate organic carbon derived from $^{210}Po/^{210}Pb$ disequilibria along the North Atlantic GEOTRACES GA01 (GEOVIDE) transect" addresses the POC export fluxes. In the second paper, we have calculated the POC fluxes using the export flux of $^{210}Po$ and the POC/$^{210}Po$ ratio in total (> 1 μm) particles and compared the estimates to those obtained using the $^{234}Th/^{238}U$ proxy.

Technical Issues:

1.  Introduction It is noted that there is significant benthic disequilibrium (210-Po deficiency) well below the euphotic zone, indeed significantly below the main thermocline at times (e.g. 4000 meters at Station 13; 1400 meters at station 60). This dilemma and benthic consequences has been discussed in the recent literature (Rigaud, et al., 2014). Page 3: As such, maybe the literature citations in the introduction that need to be updated for the current millennium!

We agree that $^{210}$Po deficits at depth can be problematic in interpretation, but others have associated them with nepheloid layers or other forms of suspended sediment near the bottom (e.g. Hu et al., 2014; Wei et al., 2014; Rigaud et al., 2015). Mid-depth deficits are more mysterious and have still not really been adequately resolved (Kim, 2001; Church et al., 2012; Rigaud et al., 2015).

We acknowledge that there were deficits in the deep waters at stations 60, and 64. However, we have almost no particle data for those depths so cannot address that in this paper.

In Fig. 2, total $^{210}$Po and total $^{210}$Pb activity were plotted as red circles and black squares, respectively. There were indeed $^{210}$Po deficits below 1400 m at station 60 but at 4000 m at station 13 there was a $^{210}$Po excess rather than a deficit. Please see the following profiles for stations 13 and 60.

[Figure]

[Figure]

We agree that the literature citations on Page 3 L55-57 need to be updated. We will add in some recent references (highlighted in bold font) as the following:

"The distribution of $^{210}$Po and $^{210}$Pb has been widely measured over the last several decades in the Atlantic (e.g. Bacon et al., 1976; Sarin et al., 1999; **Rigaud et al., 2015**), Pacific (e.g. Nozaki and Tsunogai, 1976; Murray et al., 2005; Verdeny et al., 2008), Indian (e.g. Cochran et al., 1983; Sarin et al., 1994; **Subha Anand et al., 2017**), Arctic (e.g. Moore and Smith, 1986; **He et al., 2015; Roca-Martí et al., 2016**) and Southern Oceans (e.g. Shimmield et al., 1995; Friedrich and Rutgers van der Loeff, 2002)"

2. Methods Page 4: What is meant by "Xlarge" station (26), as the number of depths are less than others?

Station naming depended on the number of casts that were conducted: XLarge stations had five casts while Super stations had > 5 casts. Because of the fewer casts, we did indeed sample at fewer depths at Station 26 than at the other stations.

3. Page 5: Is six hours sufficient for equilibration, or were there previous tests performed to verify this?' What was the time lag between sample processing on board, and nuclide separation in the lab on shore, unless both were done on board? This can be important as reviewed in Rigaud, et al. 2013. Evidently this is reflected in the data reported in

supplemental tables, although there are not errors assigned to the nuclide ratios in Table 2.

Thank you for this point. Six hours for isotope equilibration was a mistake. In fact we waited for more than 12 hours for isotopic equilibrium between $^{210}$Po and $^{209}$Po, which is recommended in RiO5 Cookbook (https://cmer.whoi.edu/wp-content/uploads/2018/01/15-Po-Pb-210-in-sewater_Co-APDC.pdf) and suggested in Rigaud et al. (2013). We will revise the manuscript accordingly.

We did our best to minimize delays in sample processing. The time elapsed between sampling and nuclide separation (first plating) was 50 and 68 days on average for the seawater samples processed at UAB and QC, respectively, and 58 and 44 days on average for the particulate samples processed at UAB and QC, respectively. This was unavoidable as the cruise was long. However, all this is taken into account into the corrections and calculations, which were performed as described in Rigaud et al. (2013).

The errors for the total particulate $^{210}$Po/$^{210}$Pb activity ratio ($^{210}$Po$_p$/$^{210}$Pb$_p$) are now added in Table 2 as follows:

Table 2. The compilation of total particulate $^{210}$Po/$^{210}$Pb activity ratios ($^{210}$Po$_p$/$^{210}$Pb$_p$) averaged in the upper 200 m, including this study.

| Region | | Sampling Method | Date | Size (μm) | Depth (m) | $^{210}$Po$_p$/$^{210}$Pb$_p$ | Reference |
|---|---|---|---|---|---|---|---|
| Arctic | CESAR | *In-situ* pump | Apr – May 83 | > 0.45 | 2-200 | 1.2 ± 0.7 | (Moore and Smith, 1986) |
| | Arctic (ARK-XXII/2) | Niskin bottle | Jul-Sep 07 | >1 | 10-200 | 0.50 ± 0.20 | (Friedrich, 2011) |
| | Chukchi Shelf | Niskin bottle | Jul-Sep 10 | > 0.45 | 0-90 | 0.37 ± 0.10 | (He et al., 2015) |
| Atlantic | F.S. Meteor | Niskin bottle | Nov-Dec 73 | > 0.4 | 0-200 | 3.1 ± 1.4 | (Bacon, 1977) |
| | Cariaco Trench | Niskin bottle | Dec 73 | > 0.4 | 0-200 | 1.4 ± 0.6 | (Bacon et al., 1980a) |
| | Labrador (R/V Knorr) | Niskin bottle | Jun 75 | > 0.4 | 0-100 | 3.9 ± 1.5 | (Bacon et al., 1980b) |
| | South of New England | Niskin bottle | Jul 80 | > 0.45 | 4-200 | 1.8 ± 0.8 | (Bacon et al., 1988) |
| | N. Atlantic (BOFS) | Niskin bottle | May-Jun 89, 90 | > 0.45 | 0-150 | 6.0 ± 4.5 | (BODC et al., 2016) |
| | South-equa. Atlantic | Niskin bottle | May-Jun 96 | > 0.7 | 10-200 | 1.3 ± 1.1 | (Sarin et al., 1999) |
| | BATS | Go-Flo bottle | Oct 96 | > 0.45 | 0-200 | 3.7 ± 3.2 | (Kim and Church, 2001) |
| | N. Atlantic (GA03) | *In-situ* pump | Oct-Nov 10, Nov-Dec 11 | > 0.8 | 30-200 | 1.5 ± 0.5 | (Rigaud et al., 2015) |
| | N. Atlantic (GA01) | *In-situ* pump | May-Jun 14 | >1 | 8-200 | 1.4 ± 0.3 | This study |
| Pacific | North Pacific | Niskin bottle | Nov 73 | > 0.4 | 10-150 | 8.5 ± 5.7 | (Bacon et al., 1976) |
| | W. Pacific (FR05/92) | Niskin bottle | Jul 92 | > 0.45 | 0-200 | 1.3 ± 1.0 | (Towler, 2003) |
| | Equa. Pacific | Go-Flo bottle | Aug-Sept 92 | > 0.45 or 0.5 | 0-200 | 5.1 ± 1.2 | (Murray et al., 2005) |
| | W. Pacific (FR08/93) | Niskin bottle | Nov 93 | > 0.45 | 0-200 | 16 ± 4 | (Towler, 2013) |
| | W. Pacific (FR07/97) | Niskin bottle | Aug 97 | > 0.45 | 0-200 | 7.2 ± 1.5 | (Peck and Smith, 2002) |
| | Aleutian Basin | Niskin bottle | Jul-Aug 08 | > 0.2 | 0-200 | 1.9 ± 3.0 | (Hu et al., 2014) |
| | E. Pacific (GP16) | *In-situ* pump | Oct-Dec 13 | >1 | 15-200 | 2.4 ± 0.6 | unpublished |
| Antarctic | S. Ocean (ANT-X/6) | Niskin bottle | Oct-Nov 92 | > 0.45 | 20-200 | 3.0 ± 1.4 | (Smetacek et al., 1997) |
| | Bellingshausen Sea | Go-Flo bottle | Nov-Dec 92 | > 0.45 | 0-100 | 14 ± 11 | (Shimmield et al., 1995) |

| | | | | | | | |
|---|---|---|---|---|---|---|---|
| | S. Ocean (ANT-XXIV/3) | Niskin bottle | Feb - Apr 08 | > 0.45 | 25-200 | 1.3 ± 0.9 | (Friedrich et al., 2011) |
| Margin Sea | S. China Sea | Go-Flo bottle | Jan-Oct 07, May 08 | > 0.45 | 0-200 | 1.7 ± 1.1 | (Wei et al., 2014) |
| | W. Taiwan | Go-Flo bottle | Apr 07 | > 0.45 | 8-25 | 0.85 ± 0.12 | (Wei et al., 2012) |
| | Yellow Sea | Niskin bottle | Feb 93 | > 0.7 | 0-100 | 0.88 ± 0.08 | (Hong et al., 1999) |
| | Mediterranean Sea | Sediment trap | Mar-Jun 03 | | 200 | 4.5 ± 1.0 | (Stewart et al., 2007) |

4.      Page 6: Who are the "Planquette group"?

Helene Planquette Group, University of Brest, co-authors in this issue.

5.      Results Page 7: As noted above, stations 13 and 60 appear to have total 210-Po
        deficiency at depth (Fig. 2), not excess.

Please see our response to Technical Issue 1.

6.      Page 8: Increase in activity with depth for both nuclides is not evident in Figs. 2 and 3,
        rather decrease.

Figures 2 and 3 are the profiles of total radionuclide activities ($^{210}Po_t$, $^{210}Pb_t$) from surface to bottom and from surface to 250 m, respectively.

In the original manuscript Page 8 lines 214-216: "The vertical profiles of $^{210}Pb_s$ were generally similar to those of $^{210}Po_s$, with relatively high activity in the surface, lower activity in the subsurface and increasing activity with depth;" $^{210}Pb_s$ and $^{210}Po_s$ refer to particulate activity in the small size fraction (not totals as in Figs 2 and 3). This data was shown only in supplementary Table S2, but we realize that it should be included in the paper and have added figures.

We will include the vertical profiles of small and large particulate radionuclide activity as the following:

[Figure]

Fig. 4. Vertical profiles of the particulate [210]Po and [210]Pb activity in the small size fraction (1-53 μm, [210]Po$_s$, [210]Pb$_s$). Note the different depth scales for the various stations and that the activity scale at Station 44 differs from the scale of all other stations. The horizontal blue line represents the bottom depth at that station.

[Figure]

Fig. 5. The vertical profiles of the particulate $^{210}$Po and $^{210}$Pb activity in the large size fraction (> 53 μm, $^{210}$Po$_l$, $^{210}$Pb$_l$) in the top 800 m. Note that the activity scale at Station 26 differs from the scale at all other stations.

Discussion Page 10: Usually in the far North Atlantic, 210-Pb association with aerosol dust is not as evident in the east, rather alternative fresh water sources (e.g. precipitation) as noted in the west.

We agree. In the submitted manuscript, we acknowledged the possible inputs of $^{210}$Pb from freshwater (e.g. sea ice processes and meteoric water) in the high latitude North Atlantic, in particular near the Greenland shelf. Nonetheless, we removed the word "unexpected" from the last sentence of this paragraph on Line 281.

7.     Page 11. The lithogenic source of a depleted 210-Po/210-Pb ratio should only be evident if the atmospheric scavenging was in the form of precipitation. Alternatively or as with lithogenic particles from the continental margin, the 210-Po has been preferentially extracted lately in fecal pellets by organisms.

We find this statement a bit confusing and are not sure how to best address it.

Near the coast, most of the lithogenic particles are terrestrial/riverine particles with a small contribution from aerosols. Aerosols have a very low $^{210}$Po/$^{210}$Pb AR (< 0.2, Baskaran, 2011) due to the short residence time of $^{210}$Pb in the atmosphere (e.g. Moore et al., 1974; Turekian et al., 1977). For the lithogenic particles sourced from land/river, the particulate $^{210}$Po depletion is more related to the nature of those particles that may preferentially adsorb $^{210}$Pb vs. $^{210}$Po as opposed to the patterns in organic materials (e.g. Fisher et al., 1983; Stewart et al., 2005).

8.     Page 12: The alternative scenario is noted here at the end of section 4.2. As such, might there be a corresponding dissolved ration greater than one?

Yes, we have looked at this relationship and will address it in much greater detail in an upcoming manuscript entirely about this topic (data presented at the Ocean Sciences Meeting, February 2018). In this study, there were a total of 13 depths where the particulate $^{210}Po/^{210}Pb$ activity ratio was lower than 1 and 8 of these depths also had a dissolved $^{210}Po/^{210}Pb$ activity ratio lower than 1.

9.     Page 13: The negative relationship between AOU and 210-Po/210-Pb is not very strong.

We agree that both negative and positive linear relationships between total particulate $^{210}Po/^{210}Pb$ AR and AOU are not very strong, with $R^2$ as 0.5 and 0.4, respectively. Nonetheless, the p-value for both linear relationships is below the significance threshold of 0.05. It appears that the negative relationship was stronger than the positive relationship in terms of $R^2$ and p-value. If we had more data available (both particulate $^{210}Po$ and $^{210}Pb$ activity and AOU), it would perhaps be possible to observe stronger relationships. This is a topic that will be explored in the future and we thought it was helpful to show the relationships while providing possible mechanisms to explain them.

10.     Page 14: Line 387 appears not to be clearly expressed indeed!

We agree and will add the following sentence to make it clearer:

"As claimed previously in Tang et al. (2017), $K_d(Po)$ is complicated because it appears to reflect both the surface adsorption and potential bioaccumulation."

11.     Conclusion Page 15: The impact of a terrestrial origin on the 210-Po/210Pb ratio less that unity might indeed be born out in the Arctic basin during summer seasons of strong biogenic processing. Maybe there is evidence in the recent GEOTRACES cruises on time scales of several months conclusive with that of the grand-daughter/parent nuclide pair?

There are two recent GEOTRACES Arctic cruises (GN01 and GN04) in 2015 which both have sampled for $^{210}Po$ and $^{210}Pb$ activity measurements. Unfortunately, the data is not yet available.

12.     Figure Captions 4) …bloom defines the date when the next bloom began.

Corrected.

13.    Figure Caption 5) The black and blue colored circles are not well distinguished.

The color code in Figure 5 is now modified. Please see below.

[Figure]

Fig. 8. Comparison of particulate $^{210}$Po/$^{210}$Pb activity ratios in the upper 200 m from this study and 20 previous studies (references in Table 2). Information about the study site, sampling date, method, and particle size of each study are shown in Table 2. The black circles represent data from previous studies while the blue circles are the results from samples analyzed at QC from three recent GEOTRACES transects (GA03, GP16, and this study, GA01 GEOVIDE). The filled magenta and open circles indicate activity ratios lower and higher than 1, respectively.

[revised manuscript text omitted]

---

## Author Comment (AC2) · 30 Jul 2018

Response to Referee 2

We thank Referee 2 for the helpful comments. We will address all changes in the revised manuscript as detailed in our responses below. The referee comments are in black and their line numbers refer to the original submitted manuscript. Our responses are in blue text.

We want to note that some of the reviewer's comments may pertain to an original draft of the manuscript which has already been revised. We have tried to address all comments, but in some cases, we do not see what the reviewer is talking about. In our response, we will only be referring to the version that is currently available on the BG website.

We also want to note that we now have submitted a companion paper to this special issue that is specifically about the particulate organic carbon (POC) export using the $^{210}Po/^{210}Pb$ technique.

**Reviewer Recommendation and Comments for Manuscript BG-2018-210**

**General comments**

The manuscript reports on total, small particles and large particles activity of 210Po and 210Pb nuclides along the North Atlantic GEOTRACES GA01 (GEOVIDE) cruise. The paper is well written, well structured and I believe that such measurements in this areas are essential for the scientific understanding of TEI's and biogenic elements in the global ocean. The approach is very good and the compilation of many other joined data (AOU, PP, SPM, chlorophyll, …) is essential to reach this goal. In addition there is a huge effort to include this new dataset with previous ones in order to get a better view of the processes controlling the behaviors of 210Po and 210Pb at a larger scale. Finally I found very interesting news findings that emerge from a new way to confront this 210Po and 210Pb dataset to other variables (comparison with chlorophyll-a from satellite-based data, AOU, …) that merit to be published.

However I found some questioning points that need to be addressed:

1.  the splitting of the samples between two different labs with two methods that differ in some points is very surprising. Some practical reasons can certainly explain this

procedure but they are not mentioned. The reader need to be sure that the results can be compared. Especially since there are distinct features that can be seen between the samples from the two labs and that a part of the discussion relies on such differences.

Thank you for your suggestion. We agree that there are some differences in the procedure between the two labs. We will add explanation in the text. Please see our responses to Specific comments 13 and 14.

2.      the last section of the discussion about the sorption, distribution coefficient and implication for particles and POC export very speculative. This is embarrassing as this appears in the abstract and the conclusion as the most important finding of the study while there are other findings much more robust that are not presented in that way.

There may be a misunderstanding, but we don't see POC export in the current abstract. We have submitted a companion paper to this special issue that is specifically about POC export, unlike this paper. We will remove the reference to POC export and rephrase the last section of the discussion to support our observations of $^{210}$Po and $^{210}$Pb distributions. Please see our response to Specific comment 47.

3.      the presentation of the context in the introduction and the state of the art about 210Po and 210Pb isotopes in the ocean is a little bit weak and I think the importance of such measurement in this area should be specifically strengthened.

Thank you for your suggestion. We will edit the Introduction section by adding more rationale for the GEOVIDE section and strengthening the objective section. Please see our responses to the Specific comments 4 and 5.

Consequently, I believe this paper must be published when these points will be addressed.

Thank you for your positive evaluation. Please see our responses to the specific comments below.

**Specific comments:**

**Title:**

1. The part of the title "partitioning between the dissolved and particles phase" is maybe not really appropriate as the most important discussions in the paper is about the processes explaining the variations in the 210Po/210Pb activity ratio within each phase (i.e., total and small/large particles).

We agree and will change the title to "Distributions of total and size-fractionated particulate $^{210}$Po and $^{210}$Pb activities along the North Atlantic GEOTRACES GA01 (GEOVIDE) transect".

**Abstract**

2.     P2, L22-23: this was not shown in the manuscript

We agree that we didn't mention it in the original manuscript.

We will add the sentence of "The average values of $K_d$(Po) was 1.6 times of those of $K_d$(Pb) in both small and total particulate phases, suggesting a higher affinity with particles for $^{210}$Po with respect to $^{210}$Pb, which is commonly observed in the global ocean (Bacon et al., 1988; Hong et al., 1999; Masqué et al., 2002; Wei et al., 2014; Tang et al., 2017)." on L399. We will keep this in the Abstract after the addition.

**Introduction**

3.     P3, L42: "seventh repetition of the OVIDE section": please precise what is the OVIDE section/program?

OVIDE is an acronym of "Observatoire de la variabilité interannuelle et décennale en Atlantique Nord," and this section covers Portugal to Greenland. This will be clearly explained in the summary paper of this special issue (Sarthou et al., in review).

4.     P3, L44-46: please give a short summary on the hydrological properties on the area.
       P3, L47-49: you should illustrate what is this expected "mixture of complex …" and why this section may present a special opportunity.

We agree that we did not provide detailed information on the hydrological properties of the study area nor the rationale for the cruise track. There will be multiple papers in this special issue and in other journals specifically describing the hydrographic and physical characteristics, justification of the GEOVIDE cruise track, and the sampling strategies (e.g. García-Ibáñez et al., 2015; Benetti et al., 2017; García-Ibáñez et al., 2018; Zunino et al., 2018; Sarthou et al., in review). We therefore will only add some information about the section as the following:

"The major goal of the international GEOTRACES program is to characterize the distributions of trace elements and isotopes (TEIs) in the ocean on a global scale, and to identify and quantify processes that control these distributions (GEOTRACES Planning Group, 2006). The GEOVIDE section was a contribution of the French GEOTRACES program to this global program in the subpolar North Atlantic. The GEOVIDE GA01 cruise was carried out in 2014 in the North Atlantic and consisted of two sections: a section along the OVIDE (Observatoire de la variabilité interannuelle et décennale en Atlantique Nord) line between Lisbon (Portugal) and Cape Farewell (southern tip of Greenland), and a Cape Farewell to St. John's (Canada) section across the Labrador Sea (Fig. 1). Since 2002, the OVIDE section has been occupied biennially to collect physical and biogeochemical data (Mercier et al., 2015). The knowledge of the currents, water masses, and biogeochemical provinces gained from the previous OVIDE campaigns enabled the optimal strategy for TEIs sampling and provided help for the interpretation of the distribution of TEIs in the subpolar North Atlantic (García-Ibáñez et al., 2015). In addition to the OVIDE line, the Labrador Sea section provided a unique opportunity to study TEIs distributions along the boundary current of the western North Atlantic subpolar gyre (Sarthou et al., in review)."

5.  P4, L72-78: I found this objectives section disappointing and clearly not ambitious enough with respect to the dataset compiled and presented in this paper. I suggest the authors to strengthen this part.

We agree with the reviewer's comments and will rewrite the objectives section as:

"In this work, we describe the distributions of total and size-fractionated particulate $^{210}$Po and $^{210}$Pb activity along the GEOVIDE cruise in the North Atlantic. These data are a significant

contribution to the high-latitude North Atlantic $^{210}$Po and $^{210}$Pb activity data set. We present a compilation of particulate $^{210}$Po/$^{210}$Pb activity ratios (AR) from previous studies in the global ocean and the results are discussed in regards to the aging of water and biochemical processes. We also describe the relationship among small particles, adsorption, and scavenging of radionuclides. These results lead to recommendations for the estimation of particulate organic carbon export flux based on the $^{210}$Po/$^{210}$Pb disequilibrium, a topic that is covered in a companion paper (Tang et al., submitted)."

**Methods**

6.      P5, L104: Please correct the sentence to avoid confusion: what was transferred into a clean bottle? The filters? The filtrate?

The filter was placed into a clean falcon tube. We will rephrase the sentence as: "Samples were filtered through a 0.45 μm membrane filter and the filters with the precipitate were placed into falcon tubes, sealed with parafilm, and stored in double-bags."

7.      P5, L107: is the "Stewart laboratory" the official name of the laboratory?

The lab doesn't have an official name. Because G. Stewart is the investigator of this lab at Queens College (QC), we used "Stewart Laboratory" in the text. We now will use QC in the revised manuscript.

8.      P5, L107-108: why this splitting procedure of the sample? The reader need to know why this splitting procedure allow to "ensure higher counting statistic in the samples". Did the laboratory performed intercalibration experiments?

We will answer the question in the text as follows: "As the delay between sample collection and first Po plating increases, the uncertainty of the calculated $^{210}$Po activity also increases. In addition, it is necessary to balance counting periods with the number of samples as the uncertainty due to alpha spectrometry counting decreases by increasing the counting time. To limit the delay between sampling and processing and to ensure higher counting statistics by having more alpha spectrometers devoted to this project, sample processing and analyses were

split between Universitat Autònoma de Barcelona (UAB) (samples from stations 1, 13, and 21) and Queens College (QC) (stations 26, 32, 38, 44, 60, 69, and 77)."

Unfortunately, there wasn't enough material to perform an intercalibration experiment.

9.      P5, L110: Please correct the sentence to avoid confusion: the filter was not evaporated to dryness.

We will change the sentence to "Briefly, the filters were digested into a solution of concentrated $HNO_3$ and HCl, and after the solution was evaporated to dryness, the samples were recovered in 1M and 0.5 M HCl solution at UAB and QC, respectively (a 0.5-2 M HCl solution is recommended, Rigaud et al., 2013)."

10.     P5, L110: Remove "eventually"

Done. Please see the previous response.

11.     P5, L112:what weak acid solution?

We now write this as "1 M/ 0.5 M HCl solution".

12.     P5, L120: write "to determine Pb recovery" instead of "to determine sample recovery".

Done.

13.     P5, L125-127: why this difference between the two labs?

The higher uncertainties for the samples processed at QC were due to additional corrections on the 1st stable Pb recovery. We will explain this in the text as:

"The activities of $^{210}Po$ and $^{210}Pb$ at the time of collection were determined by a series of corrections, including nuclide decay, ingrowth, chemical recoveries, detector backgrounds, and blank contamination following the methods in Rigaud et al. (2013). The activity uncertainties from UAB were on average 8% for both $^{210}Po$ and $^{210}Pb$ activity, while the QC uncertainties were on average 13% for $^{210}Po$ activity and 16% for $^{210}Pb$ activity. The greater uncertainties of

$^{210}$Po and $^{210}$Pb activities in the samples processed at QC were due to the longer delay between sampling and first plating (68 vs. 50 d) and higher uncertainties in the determination of the recovery of lead."

14. P6, L137: These two different digestion procedures may give different results? Please explain if tests were carried out. Are the data from the two groups comparables?

We agree that there were different digestion procedures (with or without HF), and we didn't run comparisons and need to rely on both labs working well.

15. P6, L144: what is the Planquette group?

Helene Planquette Group, University of Brest, co-authors in the special issue.

16. P6, L144: is this sentence correct: "the material on the balance of the screens and filters"?

The sentence will be rephrased as "The Helene Planquette group (University of Brest) collected subsamples from the same screens and filters that were sampled previously for radionuclides to determine major phase composition (particulate organic matter (POM), lithogenic material, calcium carbonate (CaCO$_3$), opal, Fe(OH)$_3$, and MnO$_2$) (references therein Lam et al., 2015)."

17. P6, L148-149: if the method is the same as described by Lam et al. 2015, I suggest to remove the Lemaitre et al. in prep. reference if it is not published at the time of the publication of this paper. Same comment for other reference in prep. in the manuscript.

Thank you for your suggestion. We will remove the references in prep. from the manuscript but keep the references submitted or in review as the journal suggests.

18. P6, L158: what is the Dehairs group?

Frank Dehairs group, Vrije Universiteit Brussel, co-authors in this special issue.

19.     P6, L157-164: a little bit more details is needed here: how the photometric conditions was applied on deck? I guess that 13C was spike before the incubation? …

Yes, the sample was spiked with $NaH^{13}CO_3$ before the incubation. More details on the experimental procedure were added in the text as: "The seawater was then spiked with 3 mL of a $NaH^{13}CO_3$ solution (200 mmol $L^{-1}$, 99%, Euriotop), and incubated on deck for 24 h in the circulating incubators wrapped with neutral density screens to simulate *in-situ* irradiance conditions."

20.     P7, L173-174: Before to compare the AOU data from the GEOVIDE program, you should explain how you get it. In facts, the section 2.7 is disturbing. There are two things here: the AOU and the comparison with historical data but there is no link between them. I suggest to split this section in two (even short) sections.

As suggested, we have split the original section 2.7 into two sections: section 2.7 Historical values and section 2.8 Apparent oxygen utilization as the following:

"**2.7 Historical values**

The historical data of the particulate $^{210}Po$ and $^{210}Pb$ activity, and the hydrological parameters (pressure, temperature, salinity, and dissolved oxygen) were obtained from databases and publications. The location, date, database address or publication name, and type of data (particulate $^{210}Po$ and $^{210}Pb$ activity or hydrological parameters) from all other studies is listed in supplemental Table S1.

**2.8 Apparent oxygen utilization**

Apparent oxygen utilization (AOU = $O_{2\ saturated} - O_{2\ measured}$) is defined as the difference between the saturated oxygen at a given temperature and salinity and the measured in-situ oxygen concentration (Ito et al., 2004; Duteil et al., 2013). A positive AOU indicates either water mass aging and outgassing of oxygen or biological activity, namely respiration (e.g. Keeling et al., 1998; Boyer et al., 1999). Negative AOU, indicating that the water is oversaturated with

dissolved oxygen, can appear under the conditions of an intense bloom (e.g. Coppola et al., 2017).

   The dissolved oxygen concentration was measured by Winkler titration and the saturated oxygen concentration was calculated as a function of in-situ temperature and salinity, and one atmosphere of total pressure based on the built-in function in Ocean Data View (https://odv.awi.de)."

21.   P2, L182-187: SPM, PP, chlorophyll were not considered to try to explain the 201Po-210Pb activities and activity ratios distribution?

The time-series chlorophyll-a data was considered to explain the distribution of total particulate $^{210}Po/^{210}Pb$ ratios < 1 at variable depths on L299-309 in the original manuscript.

The SPM and PP data, were indeed not used to try to explain the distribution of the radionuclide activities nor activity ratios. Instead, SPM were used to calculate the partitioning coefficient ($K_d$) while the in-situ PP and in-situ pigment data were considered to investigate the role of small particles in primary production and phytoplankton composition.

**Results**

22.   p7, L195-202: there are a clear difference between station 1, 13, 21 and the other ones. These differences also correspond to the two samples groups that were processed by two labs. This is embarrassing if there is nothing that certify that labs results can be compared.

We agree there is a clear difference between stations 1, 13, 21 and the others along the transect. We acknowledge that the greater uncertainties of $^{210}Po$ and $^{210}Pb$ activities in the samples processed at QC were due to the longer delay between sampling and first plating (68 vs. 50 d) and higher uncertainties in the determination of the recovery of lead. It is unfortunate that we could not collect additional material to perform intercalibration between the two labs.

23.    p7, L200-202: please rewrite this sentence which is very confusing.

We will change the sentence to: "$^{210}Po_t$ excesses relative to $^{210}Pb_t$, which were larger than $^{210}Po_t$ surface depletions at the same stations, were observed below the surface at some depths at stations 1, 13, and 21 in the Western European Basin (Fig. 2)."

24.    p7, L195-207: this paragraph is confusing. Please describe firstly the surface water then the depth (or in the other way) but not a mixing description.

We concur. We will rewrite section 3.1 by describing first the activity range of all samples, then the surface samples, and last the deep samples. Section 3.1 will be changed to the following:

"Total $^{210}Po$ activities ($^{210}Po_t$) in all samples ranged from 2.2 to 16.4 dpm 100 L$^{-1}$ and the mean $^{210}Po_t$ was 8.8 ± 2.4 dpm 100 L$^{-1}$ (n = 198, Fig. 2). $^{210}Po_t$ activities were generally low within the mixed layer and euphotic zone (15 – 47 m), slightly increased or remained relatively constant in the depth range between the mixed layer and 250 m, and then decreased with water depth at most of the stations except station 26. Near the seafloor, stations 1, 13 and 44 had a slight increase of $^{210}Po_t$ activity.

Total $^{210}Pb$ activities ($^{210}Pb_t$) were between 2.1 and 20.6 dpm 100L$^{-1}$ with a mean value of 10.0 ± 3.0 dpm 100 L$^{-1}$ (n = 198, Fig. 2). $^{210}Pb_t$ activities were low in the surface, slightly increased in the subsurface and decreased with water depth. Stations 1, 13, 44, and 60 exhibited an increase near the seafloor.

The mean $^{210}Po_t/^{210}Pb_t$ activity ratio (AR) of all samples was 0.92 ± 0.28 (n = 198, Fig. 2). When considering different basins separately, there is a tendency of decreasing $^{210}Po_t/^{210}Pb_t$ AR from the Western European Basin (1.10 ± 0.35) westwards to the Iceland Basin (0.90 ± 0.19) and the Irminger Sea and the Labrador Sea (0.80 ± 0.18 and 0.83 ± 0.21, respectively).

For all regions, significant deficits of $^{210}Po_t$ (0.80 ± 0.20, n = 40) were observed within the mixed layer and euphotic zone (Fig. 3). Secular equilibrium was also observed at some shallow depths (i.e. 80 m at station 44) and even in surface waters (i.e. 15 m at station 38). $^{210}Po_t$ excesses relative to $^{210}Pb_t$, which were larger than $^{210}Po_t$ surface depletions at the same stations, were observed below the surface at some depths at stations 1, 13, and 21 in the Western European Basin (Fig. 2). At depths below the surface to ~ 1500 m in the Iceland Basin, the

Irminger Sea, and the Labrador Sea, the water samples still indicated a $^{210}$Po deficiency (AR: 0.84 ± 0.17, n = 27). Secular equilibrium was generally reached near the bottom depths in all basins except at stations 13 and 60 where the water samples were either enriched in $^{210}$Po$_t$ ($^{210}$Po$_t$/$^{210}$Pb$_t$ AR = 1.58 ± 0.16) or depleted in $^{210}$Po$_t$ ($^{210}$Po$_t$/$^{210}$Pb$_t$ AR = 0.50 ± 0.12), respectively."

25.      p8, L214-216: why the figure is not shown? The particulate profiles should be plotted (at least in the appendix material).

Thank you for the suggestion. The profiles of the particulate activity in the small and large size fractions will be shown as the following:

[Figure]

Fig. 4.  Vertical profiles of particulate $^{210}$Po and $^{210}$Pb activity in the small size fraction (1-53 µm, $^{210}$Po$_s$, $^{210}$Pb$_s$). Note the different depth scales for the various stations and that the activity scale at Station 44 differs from the scale of all other stations. The horizontal blue line represents the bottom depth at that station.

[Figure]

Fig. 5.  The vertical profiles of the particulate [210]Po and [210]Pb activity in the large size fraction (> 53 μm, [210]Po_I, [210]Pb_I) in the top 800 m. Note that the activity scale at Station 26 differs from the scale at all other stations.

26.     p9, L242-244: yes, this is not surprising as the small particle are the main particulate reservoir.

Yes, we agree. No change.

27.     p9, L245-246: which particulate samples are depleted? Where they are located? In surface? Subsurface? Variable depths?

The information about those particulate samples with [210]Po/[210]Pb AR < 1 was given in Table 1 and on Line 303-304. We will add the information here as: "While the majority of particulate

matter was enriched in $^{210}$Po ($^{210}$Po$_p$/$^{210}$Pb$_p$ AR> 1), there were 13 out of 56 total samples from various depths that were depleted in $^{210}$Po relative to $^{210}$Pb."

**Discussion**

28.     p10, L264-265: large excess is not seen at depth.

In the most recent submitted draft, we don't use the word "large", but we do see $^{210}$Po activity excess at stations 1, 13 and 21 at depth. Please see the vertical profiles below (Fig. 2). In fact, the average $^{210}$Po/$^{210}$Pb AR from 100 m down to the bottom depth was 1.2 ± 0.1, 1.4 ± 0.6, and 1.1 ± 0.1 at stations 1, 13, and 21, respectively.

[Figure]

29.     p10, L260-267: I don't understand how an upwelling along the Iberian coast can bring excess 210Po all over the water column in the 3 station from the WEB.

We now rephrase the sentences on L260-267 as following:

"One possible source of these sub-surface $^{210}$Po activity excesses below 200 m at stations 1 and 13 could be the North-East Atlantic Deep Water, lower (NEADW$_L$) which was the dominant water mass in the Iberian Basin from 2000 m to the bottom, and had a concentration of silicate up to 48 µmol kg$^{-1}$ (García-Ibáñez et al., 2015). High activity of $^{210}$Po in deep samples could be due to the dissolution of diatoms or herbivore feces (Cooper, 1952). As these particles sink and dissolve, $^{210}$Po activity may have been preferentially released to the dissolved phase compared

to $^{210}$Pb activity (Bacon et al., 1976), leading to $^{210}$Po excess observed in the deep waters at stations 1 and 13. For the sub-surface $^{210}$Po activity excesses at station 1 between 400 and 1000 m where lateral inputs of particulate Fe from the margin was observed (Gourain et al., 2018), the likely process is diffusion of $^{210}$Po from those particles originated from the margin and such excess could be transported westwards to station 13 by lateral advection. An alternative source of $^{210}$Po activity excess between 50 and 250 m at stations 1 and 13 (Fig. 3) could be the eastern boundary upwelling along the coast of the Iberian Peninsula (García-Ibáñez et al., 2015). Even though no strong upwelling events were revealed from temperature and density profiles during the cruise, northerly winds favoring upwelling were recorded 2 – 3 months before the sampling (Shelley et al., 2017). The deep water may have excess $^{210}$Po activity due to the remineralization of sinking particles. The upwelling of this water mass prior to the sampling date could maintain such sub-surface excess $^{210}$Po activity. Similar findings have been reported in the Cariaco Trench for the upper 300 m of the water column by Bacon et al. (1980)."

30.    p11, L295-298: what do you mean by significant? Are they significantly different than this other station? Statistically tested? Is this confirmed from the data on the geochemical composition of SPM?

They are different from the other stations, but no statistical test was performed. Therefore, we will rephrase this sentence in the text as "In addition, the AR < 1 observed at station 1 (120, 250, and 550 m) could be associated with lithogenic particles from the Iberian Margin where 100% of the particulate Fe (PFe) had a lithogenic origin while the lithogenic contribution to PFe at other stations was smaller (Gourain et al., 2018)."

31.    p11, L304-308: this is an interesting point. Is there a figure (or a way) to illustrate this? For example a plot showing the AR in surface or subsurface as a function of the time since the last bloom?

We appreciate the suggestions. We have plotted the depths at which total particulate $^{210}$Po/$^{210}$Pb AR was found to be lower than unity as a function of the time since the last bloom in the following figure:

[Figure]

Fig. 7.  Depths at which the total particulate (> 1 µm) $^{210}$Po/$^{210}$Pb activity ratio was lower than unity vs. the time since the last bloom (data is presented in Table 1).

32.    p12, L321: Is this particulate 210Po depletion in the coastal sea related to the 210Po/210Pb AR in these the terrestrial/riverine particles or is this due to the nature of those particles that present a lower scavenging efficiency of dissolved 210Po with respect to 201Pb?

Near the coast, most of the lithogenic particles are terrestrial/riverine particles with a small contribution from aerosols. Aerosols have a very low $^{210}$Po/$^{210}$Pb AR (< 0.2, Baskaran, 2011) due to the short residence time of $^{210}$Pb in the atmosphere (e.g. Moore et al., 1974; Turekian et al., 1977). For the lithogenic particles sourced from land/river, the particulate $^{210}$Po depletion is more related to the nature of those particles that may preferentially adsorb $^{210}$Pb vs. $^{210}$Po as opposed to the patterns in organic materials (e.g. Fisher et al., 1983; Stewart et al., 2005).

33.    p12, L331: AOU must be defined in the method section. What a negative AOU value means?

Thank you for your suggestion. AOU is now defined in the method section 2.8 where the meaning of positive and negative AOU values are explained. Please see our response to specific comment 20.

34.  p12, L332: remineralization + respiration + oxidation reactions.

We have corrected it. Please see that in the following response.

35.  p12, L333-334: I do not see why water mass aging may change the OAU if there is no mineralization. To my opinion, only biogeochemical processes may change OAU values while the time can only change the intensity of O2 consumption by those biogeochemical processes. I think this should be better specified in this part to avoid confusion.

We agree that this sentence is not clear and will rephrase it as the following:

"AOU is a time-integrated measure of the amount of oxygen removed during the biogeochemical processes (e.g. respiration, remineralization, oxidation) in the ocean interior. Therefore, AOU is a product of apparent oxygen utilization rate (AOUR) and the age of water mass (e.g. Stanley et al., 2012), i.e. high AOU could be due to either intense biogeochemical processes that have occurred in a short period of time (young water mass) or weaker processes over a longer period of time (old water mass). Consequently, the rate of these biogeochemical processes and time (water mass age) would have different/similar impacts on the $^{210}Po_p/^{210}Pb_p$ AR value depending on the initial AR in the particles and the natural of the particles."

36.  p12, L336: what is an old particle? Weeks? Months? Years?

It would be months to years as after 5 half-lives of $^{210}Po$ (~ 700 days), activity of $^{210}Po$ would be 95% of the activity of $^{210}Pb$ if there is no additional removal or addition of either isotopes.

37.  p12, L336-338: time will induce an AR approaching 1: decreasing AR if the initial AR is >1 and increasing if the initial AR is < 1. Here you hypothesis that the initial AR in particle is <1 but both cases are possible. Please correct.

We agree and will change this sentence to "For example, the $^{210}Po_p/^{210}Pb_p$ AR would tend to increase with time if the initial AR is < 1 because particulate $^{210}Po$ activity would increase from the decay of $^{210}Pb$ and trend towards secular equilibrium ($^{210}Po_p/^{210}Pb_p$ AR = 1), and to decrease

with time if the initial AR is > 1 as the original excess of particulate $^{210}$Po activity would disappear after 7 half-lives of $^{210}$Po."

38.     p12, L343-357: very interesting results and interpretation! However, I have two mains questions:

Thank you for your comment. Please see our responses to the two mains questions below.

39.     Why the increase of AR from negative value to value close to 1 for OAU > 25 µmol/kg? Higher the OAU, higher the mineralization. So intuitively, the AR should be maintained more and more negative with increasing OAU?

We will rephrase the sentences on L350-355 as follows:

"The two contradictory linear trends likely reflect the nature of the particles. For example, the observation of $^{210}$Po$_p$/$^{210}$Pb$_p$ AR > 1 with AOU < 25 µmol kg$^{-1}$ may suggest relatively fresh/organic particles in the young water mass. When AOU increases either due to water mass aging or higher AOUR, the $^{210}$Po$_p$/$^{210}$Pb$_p$ AR decreases with a slope of -0.17 ± 0.04. On the other hand, refractory/lithogenic particles may be suggested by the observation of $^{210}$Po$_p$/$^{210}$Pb$_p$ AR < 1 with AOU > 25 µmol kg$^{-1}$. For those particles, increasing in AOU either due to water mass aging or higher AOUR would change the $^{210}$Po$_p$/$^{210}$Pb$_p$ AR to a much lesser degree than that for organic particles with a slope of 0.008 ± 0.003."

Increasing AOU doesn't necessarily cause the AR to decrease if the particles are lithogenic or refractory.

40.     I do not understand why it is said that this observation stands only for high latitude in the northern hemisphere. Other campaigns from high latitude in the Northern hemisphere are also reported on figure 5 but are not considered. In addition, GA-03 campaign are not from high latitude. What gives this relationship for other campaigns? Why this 4 campaigns was selected?

We found a general trend of lower particulate $^{210}$Po/$^{210}$Pb AR in samples from relatively high latitude stations in the Northern Hemisphere. We wanted to specifically study the relationship between AR and AOU for those stations. However, we couldn't obtain the hydro-data

(temperature, salinity, dissolved oxygen) from the other campaigns at the high latitude in the Northern Hemisphere, therefore AOU could not be derived as it is a function of these parameters. That is why the other campaigns from the high latitude in the Northern Hemisphere reported on Figure 5 were not considered in the AR and AOU relationship on Figure 6.

We agree that GA03 is not from high latitude. But in the original manuscript when we included GA03 into the 4 campaigns, we obtained the two-phase correlation between AR and AOU on Figure 6 (AR< 1 & AOU > 25, AR> 1 & AOU < 25). It needs to be mentioned that the two-phase correlation still exists without GA03 data but $R^2$ will decrease to ~0.3 for both relationships with less data points. We investigated the ANT-X/6 in the Southern Ocean (AOU data only available at 13 stations) as it also seemed to have relatively low AR but the similar two-phase correlation between AR and AOU did not exist.

41.    p13, L370: What do you mean by investigation of pigments? There is nothing about it in the material and methods section.

Yes, we indeed didn't include pigments in the methods section because the data has not been published yet.

We now removed the pigment data from the manuscript.

42.    p14, L377-378: what do you mean by "as the above cited papers have seen elsewhere"? Please Precise

We now removed pigment and primary production from the manuscript. Please see our response to specific comment 47.

43.    p14, L378-380: this is expected for the eastern part of the transect only?

Yes, it is. But it has been removed now. Please see our response to specific comment 41.

44.     p14, L391-392: how did you calculate the dissolved activity? This is not indicated. When you consider the Kd for the small particles you normalize with the SPM for the small particles also? Same question for the total particulate. Please precise.

We calculated the dissolved activity by subtracting particulate activity from the total activity.

Yes, we used Eq. (1) to calculate $K_d$ for both small and total particulate fractions, and $K_d$ for small and total particulate fractions was normalized by the SPM in the small and particulate fractions, respectively.

We will clarify this as following:

   "In this study, the size-fractionated data of radionuclide activity and SPM allowed us to calculate the partitioning coefficients for both radionuclides on small and total particles by using Eq. (1). The dissolved radionuclide activity was calculated as the difference between total and particulate activity. The coefficients for the small and total particulate phases were normalized by the SPM in the small and total particulate phases, respectively."

45.     p14, L399-401: How this is possible as the small particulate activity is necessary lower than the total particulate activity? Is it associated to the SPM normalization?

Yes, it is indeed associated to the SPM normalization. Please see our previous response.

46.     p14, L401-403: here you affirm that the scavenging and export is mostly driven by small particles. But there is nothing to confirm this. Although this can be plausible, this is just an hypothesis.

We agree that we don't have direct evidence of small particles sinking nor that export was driven by the small particles. We therefore removed this topic from this manuscript and will discuss it further in the companion paper. However, the comparison of $K_d$ for both radionuclides in the small size fraction vs. total size fraction suggests that the adsorption/scavenging of radionuclides were driven by the small particles.

47.     p12-14, L362-404: this section is very surprising. From the title of the section I excepted to find POC export calculation. In facts, there is no data really discussed or even showed

(pigment, primary production, …) and most of the discussion is based on hypothesis without real solid basis to support them. I suggest to rewrite this section around concrete data only and to change the title of this section.

In fact, we have submitted two manuscripts to this special issue. In the manuscript reviewed here we discuss the general distribution of $^{210}$Po and $^{210}$Pb activity along the GEOVIDE transect. The second manuscript entitled "The export flux of particulate organic carbon derived from $^{210}$Po/$^{210}$Pb disequilibria along the North Atlantic GEOTRACES GA01 (GEOVIDE) transect" addresses the POC export fluxes. In the second paper, we have calculated the POC fluxes using the export flux of $^{210}$Po and the POC/$^{210}$Po ratio in total (> 1 μm) particles and compared the estimates to those obtained using the $^{234}$Th/$^{238}$U proxy.

We agree that the title of this section is not appropriate. We will change it to "Relationship among small particles, adsorption, and scavenging".

We now removed the statements about the pigment and primary production from this section and will rewrite it as follows:

**"4.4 Relationship among small particles, adsorption, and scavenging**

[revised manuscript text omitted]

**Conclusion:**

48.     p15, L415-420: again this was not clearly demonstrated. This conclusion should be very robust because it can have large implications in the future sampling strategy. Differently: does the sampling and analysis of two particulate size fractions is necessary in the future? So this has to be very robustly demonstrated. I agree with the fact that the high proportion of particulate nuclides is found in the small particle indicates that small particles are important in the sorption process. But I'm clearly not convinced from the data showed in the manuscript there is evidence to say that the small particles play an important role in the export of particles. If so, this should be strengthened. I may suggest to synthesis the most important findings based on the data only. There is nothing on the time elapsed since the last bloom for example.

We agree that conclusions should be based on the data only and, therefore, we will remove the text about the relationship between small phytoplankton and export. Conclusions will be changed to the following:

"In this study, we reported the vertical distribution of total and size-fractionated particulate $^{210}$Po and $^{210}$Pb activities in the North Atlantic during the GEOVIDE GA01 cruise. More than 90% of the radionuclide activity was found in the dissolved phase, while a small proportion was associated with particles in this transect. Total $^{210}$Po activity was generally depleted relative to total $^{210}$Pb activity in the upper 100 m due to the preferential adsorption of $^{210}$Po activity by particles. Such deficiencies of $^{210}$Po activities generally extended to the deep waters at most of the stations. In the Western European Basin, the excess of $^{210}$Po activities at stations 1 and 13 in the North East Atlantic Deep Water was attributed to the release of $^{210}$Po during dissolution of sinking biogenic particles.

There appear to be geographic differences in particulate $^{210}$Po/$^{210}$Pb activity ratios measured during GEOVIDE and previous studies, with particularly low values in the high-latitude North Atlantic and Arctic. While this observation deserves more attention, we support previous suggestions that this is due to the terrestrial origin/riverine input of particles with a low $^{210}$Po/$^{210}$Pb AR into the river-dominated shallow seas of the Arctic. The age of the particles and water masses as well as the importance of biogeochemical processes (e.g. respiration, remineralization) may also explain some of these observations, as there was a significant relationship between the total particulate activity ratio and AOU when both were measured in the North Atlantic (> 20 ºN) and Arctic Oceans.

Over 80% of the particulate radionuclide activity was on small particles, indicating that the scavenging of both radionuclides was driven by small particles. Therefore, we suggest considering the activities of $^{210}$Po and $^{210}$Pb from both small and large particles in order to study the water column $^{210}$Po/$^{210}$Pb disequilibria and quantify POC export along the GA01 transect. This has been addressed in a companion paper in this issue. We recommend that both small and large particles should be sampled for POC/$^{210}$Po estimates for the application of the $^{210}$Po/$^{210}$Pb method in future studies of POC export."

**Fig 3:**

49.     I doubt the sentence "A closer look at only the zoom" is correct in the caption

Those words will be removed from the caption. The caption will be changed as: "The upper 250 m of the depth profiles of total $^{210}$Po ($^{210}$Po$_t$, red circles) and $^{210}$Pb activities ($^{210}$Pb$_t$, grey squares) along the GEOVIDE section. The horizontal orange and magenta lines denote the mixed layer depth (MLD) and the base of the euphotic zone ($Z_{1\%}$), respectively. The depth profiles are shown in the order of sampling and grouped by region (refer to Fig. 2 for the text abbreviations)."

50.     Stn 60: 2 dot are missing for 210Pb at approximatively 50 m and 120 m depth.

The range of horizontal axis (activity) for each plot will be changed from 0-20 to 0-25 dpm 100 L$^{-1}$ and the two data points will be shown in the plot of stn. 60. Please see the plot below:

[Figure]

**Figure 6:**

51.     negative AOU value need to be explained?

Negative AOU value is now explained in the text. Please see our response to specific comment 20.

52.     with the uncertainty on Po/Pb AR there is (most of the time) not significant deviation from the 1 AR for the "other points". I suggest to integrate the "other points" within the regression keeping the only separation lower or above 25 µmol/kg for OAU.

We agree. The original Figure 6 will be changed to the following:

[Figure]

Fig. 9. The relationship between AOU ($\mu$mol kg$^{-1}$) and total particulate $^{210}$Po/$^{210}$Pb activity ratio ($^{210}$Po$_p$/$^{210}$Pb$_p$) from the upper 200 m in the northern hemisphere (> 22 ºN) investigated by a linear regression model (red and blue lines). The 40 stations include data from previous studies, ARK-XXII/2 (77.38-87.83 ºN, n = 15) in the Arctic, BOFS (48.89-49.87 ºN, n = 7), GA03 (22.38-39.70 ºN, n = 7), and this study, GA01 (40.33-59.80 ºN, n = 11) in the North Atlantic. The horizontal dashed line represents $^{210}$Po$_p$/$^{210}$Pb$_p$ AR = 1 and the vertical dashed line represents AOU = 25 $\mu$mol kg$^{-1}$. Blue circles denote AOU < 25 $\mu$mol kg$^{-1}$, while red circles denote AOU > 25 $\mu$mol kg$^{-1}$.

**Figure 7:**

53.     the axis labels on the figure and in the caption are not the same. Please homegeneize.

We agree. The caption of Figure 10  (no longer 7 because of the addition of figures) will be changed as:

[revised manuscript text omitted]